# A multiomic characterization of the leukemia cell line REH using short- and long-read sequencing

Mariya Lysenkova Wiklander[1,2] , Gustav Arvidsson[1,2] , Ignas Bunikis[2,3,4] , Anders Lundmark[1,2], Amanda Raine[1,2,4] , Yanara Marincevic-Zuniga[1,2,4] , Henrik Gezelius[1,2,4] , Anna Bremer[2,3,5], Lars Feuk[2,3,4] , Adam Ameur[2,3,4] , Jessica Nordlund[1,2,4]

The B-cell acute lymphoblastic leukemia (ALL) cell line REH, with the t(12;21) *ETV6::RUNX1* translocation, is known to have a complex karyotype defined by a series of large-scale chromosomal rearrangements. Taken from a 15-yr-old at relapse, the cell line offers a practical model for the study of pediatric B-ALL. In recent years, short- and long-read DNA and RNA sequencing have emerged as a complement to karyotyping techniques in the resolution of structural variants in an oncological context. Here, we explore the integration of long-read PacBio and Oxford Nanopore whole-genome sequencing, IsoSeq RNA sequencing, and short-read Illumina sequencing to create a detailed genomic and transcriptomic characterization of the REH cell line. Whole-genome sequencing clarified the molecular traits of disrupted ALL-associated genes including *CDKN2A*, *PAX5*, *BTG1*, *VPREB1*, and *TBL1XR1*, as well as the glucocorticoid receptor *NR3C1*. Meanwhile, transcriptome sequencing identified seven fusion genes within the genomic breakpoints. Together, our extensive whole-genome investigation makes high-quality open-source data available to the leukemia genomics community.

## Introduction

REH, established in 1974 as the first B-cell precursor acute lymphoblastic leukemia (ALL) cell line, was derived from the peripheral blood of a 15-yr-old female patient at relapse (Rosenfeld et al, 1977). REH has a near-diploid karyotype characterized by structural rearrangements, including several interchromosomal translocations. Among the features described in this cell line is a four-way rearrangement t(4;12;21;16) that results in the canonical subtype-defining *ETV6:RUNX1* fusion gene (Uphoff et al, 1997) and glucocorticoid (GC) resistance attributed to the lack of a GC receptor (Bachmann et al, 2007). Although this cell line is used extensively in

leukemia research (Tzoneva et al, 2018; Cousins et al, 2022; Leo et al, 2022; Wray et al, 2022), its complex karyotype has never been comprehensively characterized.

Traditional karyotyping and banding techniques struggle to resolve structural variants under 5–10 Mb in size, whereas the resolution of FISH and microarray analyses can go down to ~150 kb (Rassekh et al, 2008; Dwivedi et al, 2014). Combined whole-genome and transcriptome sequencing (WGTS), capable of more accurately resolving clinically relevant structural variants (SVs) and fusion genes than conventional techniques, is being increasingly explored as a diagnostic tool to detect such aberrations in a clinical setting (Tsang et al, 2021; Berglund et al, 2022; Cuppen et al, 2022; Ryan et al, 2023). Until recently, clinical variant detection using WGTS has been predominantly assessed using short-read next-generation sequencing because of its lower cost, higher accuracy, and more extensively validated computational pipelines compared with long-read technologies (Amarasinghe et al, 2020; Xiao et al, 2021; Jobanputra et al, 2022). However, inherent challenges exist with short-read sequencing, which include difficulty in mapping short reads that arise from repetitive sequences, sequencing through regions of high GC content, and detecting large-scale SVs (Roberts et al, 2021; Meggendorfer et al, 2022; Olson et al, 2023). The "dark" regions of the human genome have been difficult to assemble and are thus poorly covered by standard short-read sequencing, preventing the identification of potential disease-relevant SVs (Ebbert et al, 2019). Long-read sequencing technologies, including those developed by Pacific Biosciences (PacBio) and Oxford Nanopore Technologies (ONT), address these issues with their capability to read through repetitive regions with variable GC content, and to resolve single-nucleotide variants (SNVs) and SVs even in the most challenging genomic regions (Ardui et al, 2018; Amarasinghe et al, 2020; Rausch et al, 2023). Long reads have shown superiority to short reads for SV detection particularly when used with multiple SV callers (Liu et al, 2020). Importantly, long-read technologies have made remarkable improvements in throughput and base calling

---

[1]Department of Medical Sciences, Uppsala University, Uppsala, Sweden  [2]SciLifeLab, Uppsala University, Uppsala, Sweden  [3]Department of Immunology, Genetics and Pathology, Uppsala University, Uppsala, Sweden  [4]National Genomics Infrastructure, Uppsala University, Uppsala, Sweden  [5]Department of Clinical Genetics, Uppsala University Hospital, Uppsala, Sweden

Correspondence: jessica.nordlund@medsci.uu.se

accuracy, which is now over 99% (Conlin et al, 2022), making them an interesting alternative for future clinical cancer diagnostics.

There remain numerous clinically important genomic regions overlooked by short-read next-generation sequencing that have the potential to be resolved by long reads. In the present study, we explore the utility of short- and long-read sequencing techniques to produce a detailed genomic and transcriptomic characterization of REH, providing a resource for future leukemia research using this important and highly complex cell line. In addition, we compare the performance of the different sequencing technologies and analytical pipelines used to produce this high-resolution reference dataset.

# Results

## Overview of the genome

The human pre–B-cell ALL cell line REH was obtained from DSMZ (ACC 22), cultured according to the supplier's specifications, and authenticated using STR analysis. G-banded karyotyping was then used to verify the presence of the chromosomal features described previously in literature (Uphoff et al, 1997) and by the cell line vendor (DSMZ) (Fig 1A). Karyotyping confirmed most of the large chromosomal aberrations expected in REH: a deletion on the short arm of chromosome 3, trisomy 16, the balanced t(5;12), and monosomy X. It did not, however, completely resolve the four-way t(4;12;21;16). A comparison of the features resolved in the different karyotypes is available in Table S1.

The REH cells were subjected to genomic and transcriptomic sequencing. Of 33 datasets generated (Lysenkova Wiklander et al, 2023), the present study used three whole-genome sequencing (WGS) and two RNA-seq datasets (Table 1). The Illumina WGS dataset had an average depth of coverage of 34 reads, whereas ONT had 18 and PacBio 15; the Illumina WGS was sequenced PE-150 bp, while the ONT and PacBio median read lengths were 13,261 bp and 23,761 bp, respectively (Fig 1B). The IsoSeq median read length was 3,747 bp (Fig S1), whereas the Illumina RNA-seq was sequenced PE-100 bp. All sequencing datasets had an error rate of less than one percent, except for ONT (6.55%). Further sequencing statistics of these datasets are detailed in Table S2.

Three sets of SNV calls were generated by running DeepVariant on the WGS datasets. The variant allele frequency of the SNVs was used to assess uniparental disomies and other potential losses of heterozygosity (LOH) in the cell line, using the SNV callsets generated from Illumina (Fig 1C), ONT (Fig S2A), and PacBio (Fig S2B) data. In addition to confirming the del(3), +16, and −X, a partial trisomy of chromosome 21 was detected, suggesting two copies of the der(16) chromosome resulting from the t(4;12;21;16). Copy neutral loss of heterozygosity (cnLOH) was also detected across much of the short arm of chromosome 9, stretching from p13.2 (between exons 4 and 5 of RNF38) to the p-terminal.

Three SV callsets were created by running TIDDIT on Illumina short-read WGS data and Sniffles on PacBio and ONT long-read WGS data, from which consensus sets were subsequently generated using the SURVIVOR algorithm (Jeffares et al, 2017). To rule out the presence of subclones, the allele fractions of the three-way consensus callset, containing the SVs called in all three WGS datasets, were plotted to ensure that they follow a binomial distribution (Fig S3).

Two haplotype-aware de novo assemblies were generated to provide chromosomal context for SVs. The first assembly was created using PacBio circular consensus sequencing (CCS) reads and hifiasm (Cheng et al, 2021) with no pre-processing or polishing. The second assembly was created using ONT ultralong reads, Flye (Kolmogorov et al, 2019), and Medaka polishing. The hifiasm assembly generated 2,580 contigs, N50 of 3.5 Mb and L50 of 255, which was outperformed by the Flye assembly (2,060 contigs, N50 of 58 Mb and L50 of 20). The Flye assembly was selected for phasing analysis.

## The SV landscape stratified by size

In total, TIDDIT generated 5,726 unfiltered SV candidates from the Illumina data, whereas Sniffles generated 36,121 candidates from the PacBio and 36,648 candidates from the ONT data. The distribution of these SVs over chromosomal loci was assessed with heatmap visualizations of the long-read consensus callset (Fig 2A), the TIDDIT callset (Fig S4A), the PacBio and ONT Sniffles callsets (Figs S4B and S4C), and the three-way consensus callset (Fig S4D). Of SVs with length > 100 bp, there were 10,262 variants in the long-read consensus set (called in both ONT and PacBio), whereas the three-way consensus set (intersection of all three WGS callsets) contained 2,072 variants. The non-translocation SVs were binned by size into Small (100 bp–1 kb), Medium (1–10 kb), and Large (>10 kb); in these categories, the long-read consensus sets contained 8,993, 1,420, and 19 variants, respectively, whereas the three-way consensus sets contained 1,485, 414, and 34 variants (Fig 2B). The frequencies of SVs stratified by size were visualized using strip plots (Fig 2C).

The original Sniffles and TIDDIT callsets were then programmatically filtered (see the Materials and Methods section). After automated filtering for large-scale SVs (>100 kb) and interchromosomal translocations, there remained 694 SV candidates from Illumina, 127 from PacBio, and 35 from ONT. All remaining Sniffles candidates were manually inspected in IGV, whereas a pseudo-random sampling from the TIDDIT set selected 29 candidates across 16 chromosomes for further inspection.

## Refinement of known REH aberrations at base-pair resolution

The established and expected chromosomal aberrations were detected in each of the three SV callsets (Table 2). These features include a deleted chromosome X, a gain of chromosome 16, a 26-Mb del(3)(p22.3p14.2), a balanced translocation between chromosomes 5 and 12, and, finally, a four-way rearrangement between chromosomes 4, 12, 21, and 16, resulting in the ETV6::RUNX1 fusion gene. The t(5;12) and the four-way translocation were determined to involve two different homologs of chromosome 12, as the two p-arms of chromosome 12 (from p13.2 to the p-terminal) were found to be fused to either chromosome 5 or chromosome 21.

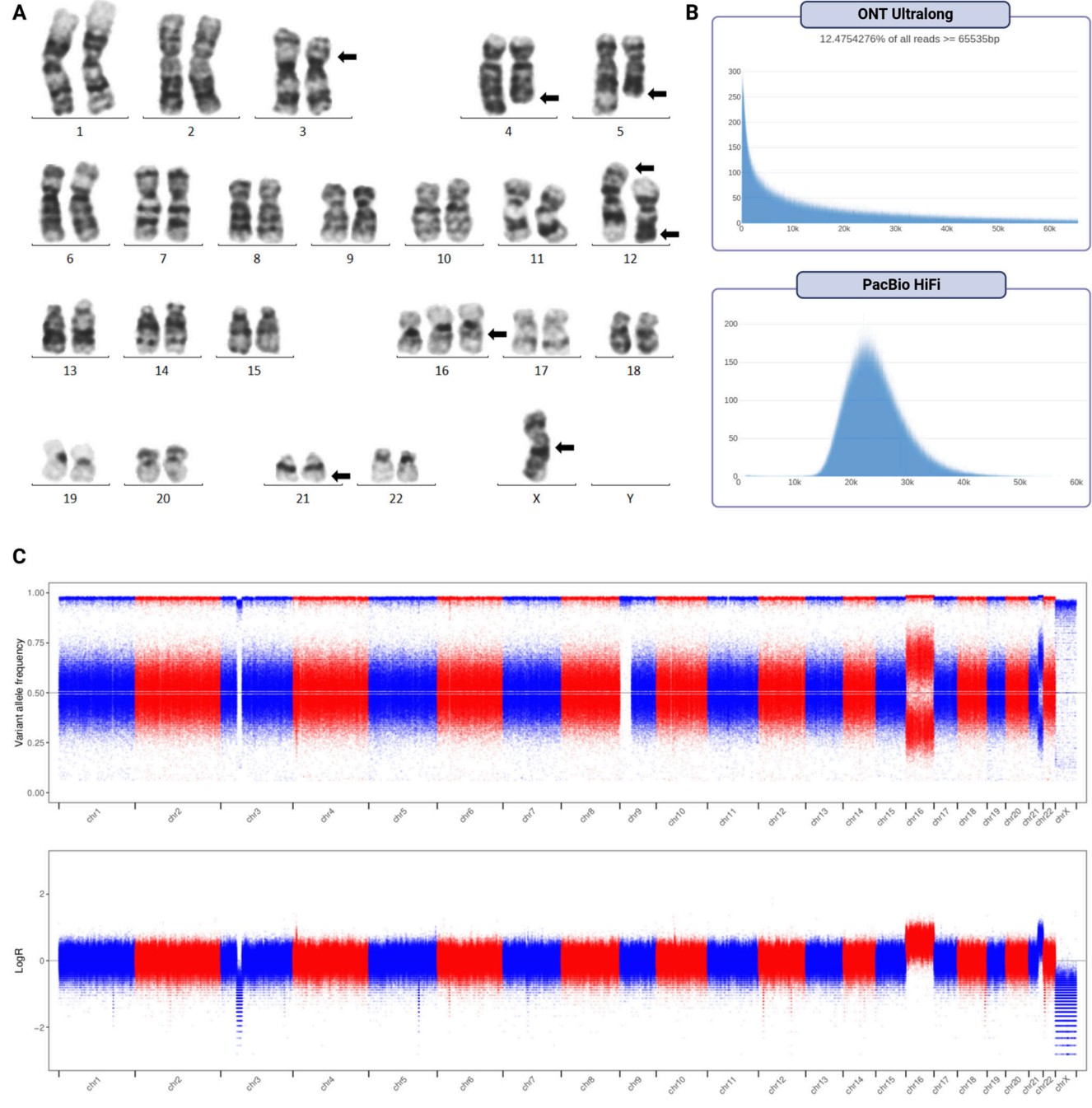

**Figure 1. Overview of the REH cell line.**
**(A)** G-banded karyotyping used to verify the REH karyotype provided by the cell line vendor. Arrows mark the chromosomes with visible aberrations, reflecting the major features of the stemline described in the DSMZ karyotype: *46(44-47)<2n>X, -X, +16, del(3)(p22), t(4;12;21;16)(q32;p13;q22;q24.3)-inv(12)(p13q22), t(5;12)(q31-q32;p12), der(16)t(16;21)(q24.3;q22)—sideline with inv(5)der(5)(p15q31),+18*. G-banded karyotyping showed that the cells in the present study did not contain the sideline. **(B)** Read length distributions of the long-read whole-genome sequencing datasets. **(C)** Variant allele frequencies of the single-nucleotide variants called by DeepVariant in the Illumina whole-genome sequencing data. The allele fractions of these single-nucleotide variants, relative to the reference alleles, are binomially distributed, with 1.0 indicating homozygous variants and a mean of 0.5 indicating heterozygous variants.

The molecular events of the four-way translocation were reconstructed as follows. The p-arm of chromosome 12 (12 Mb), with a breakpoint at p13.2 in intron 5 of *ETV6*, was translocated to chr21q22.12, resulting in the canonical subtype-defining fusion gene *ETV6::RUNX1*. The q-arm of chromosome 21 (12 Mb), with a breakpoint at q22.12 in intron 1 of *RUNX1*, was translocated to chr16q24.3. The q-arm of chromosome 16 (271 kb), with a breakpoint at q24.3 in intron 4 of *PRDM7*, was translocated to chr4q32.1. Finally, the q-arm of chromosome 4 (30 Mb), with a breakpoint at q32.1, was translocated to chr12q23.1 (where chromosome 12 had undergone an 85-Mb

**Table 1. Overview of the REH sequencing and analysis methods.**

| | | Short-read (Illumina) | Long-read (PacBio) | Long-read (ONT) |
|---|---|---|---|---|
| WGS | Library preparation | TruSeq DNA PCR-Free | SMRTbell | SQK-LSK110 Ligation Sequencing Kit |
| | Instrument | HiSeq X 10 | Sequel II | PromethION 24 |
| | Raw reads (GB) | 72 | 0.7 (CLR) | 55 |
| | | | 45.2 (HiFi) | |
| | Median coverage | 34 reads/17 read pairs | 15 | 18 |
| | Median read length | PE-150 bp | 23,761 bp | 13,261 bp |
| | Analysis tools | DeepVariant, Mutect2, Sarek, TIDDIT | DeepVariant, pbmm2, Sniffles2 | DeepVariant, Minimap2, Sniffles2, Flye, Hapdup, Hapdiff |
| RNA-seq | Library preparation | TruSeq Stranded Total RNA | SMRTbell IsoSeq | |
| | Instrument | NovaSeq 6000 | Sequel II | |
| | Raw reads (GB) | 4 | 4.9 (standard-length) | |
| | | | 9.1 (full-length) | |
| | Median read length | PE-100 bp | 3,747 bp | |
| | Analysis tools | nf-core/rnafusion | Minimap2, cDNA_Cupcake, JAFFAL | |

inversion between p13.2 and q23.1), completing the reciprocal exchange.

Combining the different SV callsets facilitated the resolution of these breakpoints down to base-pair accuracy, thereby resolving three inconsistencies in the previously documented REH karyotypes. First, each of the three callsets resolved the breakpoints of the balanced t(5;12) to 5q23.2-23.3, with a 2.2-Mb deletion on chr5q23 occurring between the breakpoints. The previous karyotypes each erroneously placed the breakpoint on chromosome 5 at q31-q32. Second, the breakpoints of the del(3) were unclear from the karyotypes, ranging from p13-p22 to p21.3-p24, or missing altogether. The SV callsets unanimously resolved this deletion to p14.2-p22.3. Finally, the karyotypes showed a discrepancy in the endpoint of the inv(12)(p13), placing it at either q22 or q23; the callsets resolved its location to q23.1 (Table S1).

Of note, the DSMZ karyotype documents the presence of a sideline containing an extra chromosome 18 and an inversion at chromosome 5. Neither of these aberrations were detected by our STR analysis or G-banding, indicating that the REH cells analyzed by us did not contain this sideline. These aberrations are also absent from the karyotype generated by Uphoff et al (1997).

Known aneuploidies were confirmed in the WGS data: a deletion of chromosome X was detected in all datasets by a marked decrease in the average depth of coverage (DC), whereas an additional chromosome 16 was detected by a corresponding increase (Table S3). The two copies of derived chromosome 16 resulting from the translocation t(16;21) were confirmed through DC and analysis of reads mapping to both chromosomes 16 and 21. The breakpoint was supported by 18 ONT split reads (average ONT DC chr16 = 26.4 versus 18.2 for diploid chromosomes), 19 PB split reads (average PB DC chr16 = 20.8 versus 14.6 for diploid chromosomes), and 32 Illumina discordant read pairs (average

Illumina DC chr16 = 51.5 versus 34.7 for diploid chromosomes). A single set of breakends was found for this rearrangement (chr21: 34947932, chr16:90067326), suggesting that this aneuploidy was caused by an aberrant mitotic event occurring after the translocation.

### Large-scale SVs discovered in the REH genome

In total, 15 intrachromosomal and eight interchromosomal SVs were found (Fig 3), of which one interchromosomal translocation and 14 intrachromosomal SVs (deletions, duplications, and inversions >100 kb) were previously unidentified in the karyotypes (Table 2). A rearrangement was identified involving a 1.4-Mb segment of chromosome 2 inverted at p11.2, 58 kb of which was inserted into chromosome 1, resulting in derived chromosome der(1) inv(2)(p11.2p11.2)ins(1;2)(q21.1;p11.2). Our findings also include a 116-kb duplication on chromosome 1p21.1, a 1.1-Mb inversion on chromosome 16p12.2, and deletions on eight different chromosomes. These include a deletion on chromosome 3 at q26.32 (146 kb), deleting exons 2–5 of the gene *TBL1XR1*, which has been implicated in GC drug resistance (Jones et al, 2014) (Fig S5). A deletion on chromosome 5q31.3 (205 kb) was found to focally delete exons 2–9 of the GC receptor gene *NR3C1* (Xiao et al, 2019) (Fig S6). Meanwhile, LOH on the p-arm of chromosome 9 leading to the homozygosity of a 24.5-Mb deletion at p21.3 was shown to delete both alleles of *CDKN2A*, which may be associated with inferior outcome (Kathiravan et al, 2019). A deletion on chromosome 12q21.33 (260 kb) was found in the region encoding the ALL-associated gene *BTG1* (van Galen et al, 2010) (Fig S7), whereas three deletions were found on chromosome 18, one at q21.1 and two at q23, including the 132-kb deletion affecting exon 10 of the leukemia-associated gene *NFATC1* (Medyouf & Ghysdael, 2008) (Fig

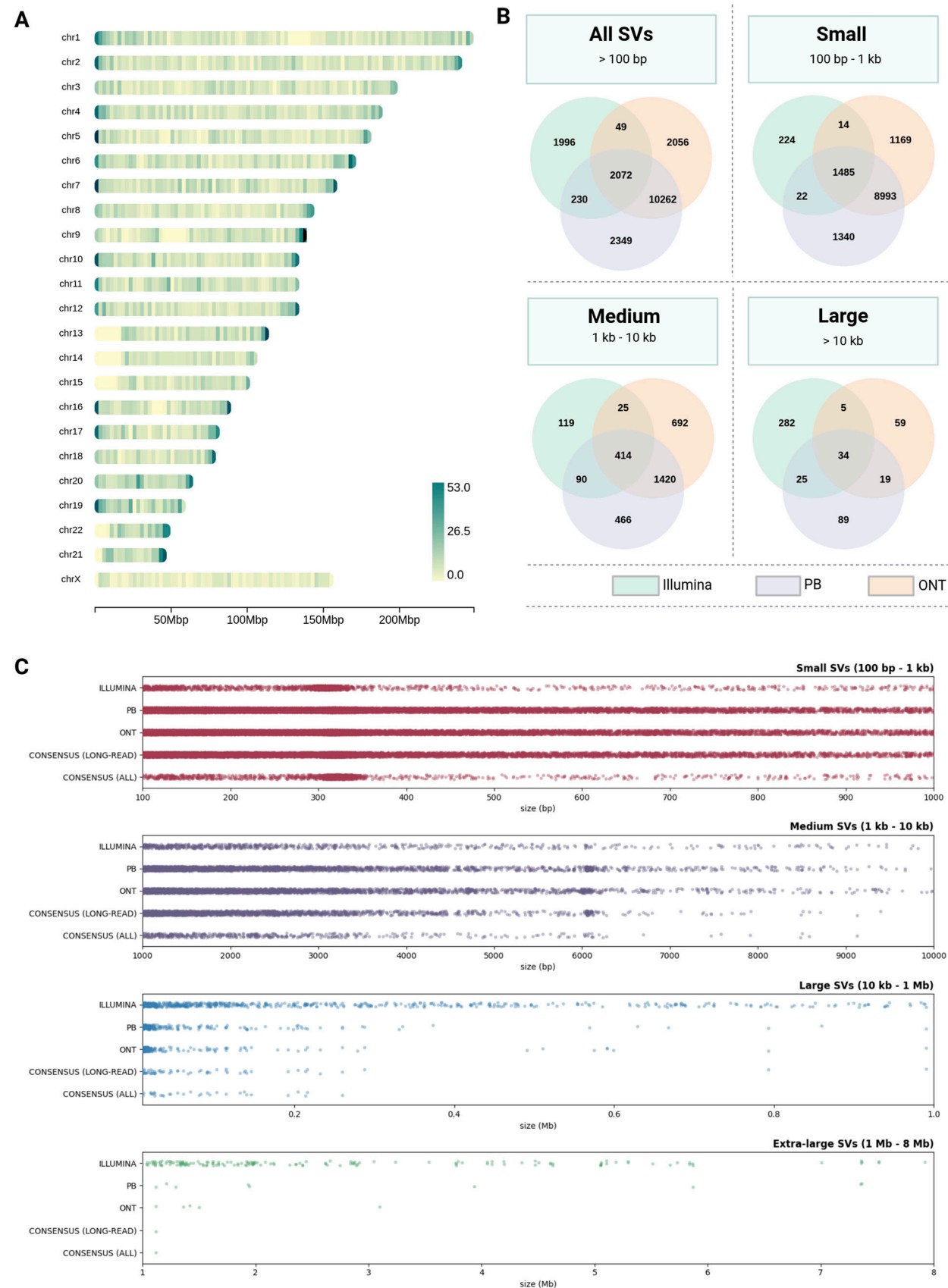

S8). Finally, two deletions were discovered on chromosome 22q11.22, of which a 214-kb variant was found to entirely delete the gene *VPREB1*, which encodes the surrogate light chain involved in the formation of the pre–B-cell receptor (pre-BCR) and whose genetic loss contributes to leukemogenesis (Mangum et al, 2014) (Fig S9).

Of the 23 confirmed SVs, a total of 19 were called in all three SV callsets, including the eight aberrations known from the karyotypes and 11 of the novel SVs. An additional two SVs, in repetitive regions of chromosomes 1 and 2, were called in both PacBio and ONT, but not in Illumina data, whereas one SV was called in both Illumina and ONT. Finally, the homozygous deletion in chromosome 9 was only detected in Illumina data. All of the confirmed SVs were supported by at least three split reads from both ONT and PacBio, and at least three discordant read pairs from Illumina, with an average read support of 9.6 from PacBio, 11.2 from ONT, and 15.9 from Illumina (Table 2).

We evaluated the strengths and weaknesses of short- and long-read sequencing technologies in the context of characterizing the REH cell line. Overall, Sniffles run on ultralong ONT reads outperformed in the discovery of large-scale rearrangements, showing high sensitivity and a low false-positive rate relative to the other datasets. Sniffles/ONT called 22 of the 23 confirmed interchromosomal or large-scale SVs > 100 kb (95.65% sensitivity), whereas both Sniffles/PacBio and TIDDIT/Illumina called 21 (91.3%). The Sniffles/ONT callset contained the smallest number of false positives (128; 85.33% FPR), which were fewer than the Sniffles/PacBio callset (606; 96.65% FPR) and TIDDIT/Illumina callset (1,469; 98.59% FPR) (Fig S10A; Table S4).

The TIDDIT callset had a high sensitivity rate, detecting all of the confirmed SVs except for those occurring in highly repetitive regions. It was also the only dataset to detect the homozygous deletion on chromosome 9, despite high coverage of the breakpoints with long reads. However, it suffered from a high false-positive rate. Of the 29 SV candidates randomly sampled from the filtered TIDDIT callset, none were visibly supported by long reads in IGV; these SV candidates were called in regions containing indels >10 bp, which were resolved in the long-read data but mismapped and misidentified as translocations or other SVs in the Illumina data.

PacBio data, which had lower average DC than ONT (15 PB versus 18 ONT), did not detect any SVs that were missed in the other two callsets; however, in both the PB and ONT datasets, Sniffles detected an ins(1;2) and a 116-kb dup(1), both missed in the Illumina data and both occurring in highly repetitive regions indicated by spikes in DC (Fig 3).

The long-read technologies facilitated the confirmation of large-scale aberrations using split-read analysis. The visualization tool Ribbon was used to visualize reads spanning complex breakpoints such as the one between chromosomes 12, 16, and 21, and to explore how specific split reads map to the reference genome (Fig S10B). The ultralong ONT reads in particular helped resolve the rearrangement

der(1)inv(2)(p11.2p11.2)ins(1;2)(q21.1;p11.2), where several ultralong reads spanned the entire 58-kb translocated region of chromosome 2 and its flanking regions in the derivative chromosome 1 (Fig S10C).

Phasing of the large-scale SVs was performed by creating a de novo assembly of the ONT reads with Flye and haplotyping with Hapdup, followed by alignment of the haplotypes to the reference genome and SV calling using Hapdiff. One can infer whether a pair of SVs occurs on the same homolog if they occur on the same contig or, alternatively, if a single contig covers the breakpoints of both events and contains neither of the SVs. The three large-scale deletions on the q-arm of chromosome 18 were mapped to a single contig, indicating that they all occur on the same homolog. However, this approach struggled to properly resolve rearrangements with higher complexity; for example, it mapped the large-scale inv(12) to the same homolog as the t(5;12), contradicting cytogenetic evidence. Furthermore, of the 31 SV breakpoints that were assigned to a haplotype, 14 of the contigs containing heterozygous SVs (45%) were erroneously assigned to both maternal and paternal haplotypes, whereas Hapdiff only called 10 of the 24 confirmed large-scale SVs (Table S5).

## Seven expressed fusion genes, including two in-frame fusions

Short- and long-read RNA-sequencing data were used together with seven fusion gene callers to detect potential fusion genes in REH. In total, the unfiltered fusion callset consisted of 11,099 fusion gene candidates. After programmatic filtering, 30 candidates remained. Additional manual examination of split and spanning reads, as well as discordant mate pairs, retained seven high-confidence fusion candidates that could be confirmed in IGV with supporting genomic breakpoints in the WGS data (Fig 4).

Of the seven confirmed fusion genes, two were in-frame. First, the expected *ETV6::RUNX1* fusion, resulting from the canonical subtype-defining translocation t(12;21), was detected as two splicing variants. Transcripts of both variants retained the sterile alpha motif/pointed domain from ETV6, and the Runt and Runx inhibition domains from RUNX1. *ETV6::RUNX1* was detected in both the Illumina RNA-seq and IsoSeq datasets (five and 62 supporting reads, respectively) with support from five fusion callers. The two splice variants of *ETV6::RUNX1* were in line with the genomic breakpoints found in intron 5 of *ETV6* and intron 1 of *RUNX1*; however, in the most prevalent *ETV6::RUNX1* transcript, exon 5 of *ETV6* was spliced to exon 2 of *RUNX1* (Fig 4A).

Five splicing variants of a second in-frame fusion gene, *RUNX1::PRDM7*, resulted from the t(16;21) occurring in two copies of the der(16). In the splicing variant involving exon 5 of *PRDM7*, the *RUNX1::PRDM7* fusion retained the promoter from RUNX1 and the SSXRD motif from PRDM7. *RUNX1::PRDM7* was highly expressed, detected by all seven fusion callers, and supported by 144 Illumina reads and 24 IsoSeq reads. The genomic breakpoint in *PRDM7* was in

---

**Figure 2. Structural variants detected in PacBio, ONT, and Illumina whole-genome sequencing data.**
**(A)** Chromosomal heatmap of the long-read consensus callset, showing the total number of structural variant (SV) calls at each locus that were detected in both PacBio and ONT data. **(B)** Venn diagram showing the number of SV calls found to overlap in each combination of callsets. **(C)** Strip plots showing SVs from each callset, stratified by size, with the bottom two strips in each section visualizing the long-read consensus callset and the three-way consensus callset, respectively.

**Table 2. Structural variants >100 kb observed in REH across the different SV callsets.**

| Status | Called in | SV info | | | | | | Read support[a] | | | Genes of interest |
|---|---|---|---|---|---|---|---|---|---|---|---|
| | | Type | Chrom | Locus | Start | End | Size | PB | ONT | Ilmn | |
| Known REH SVs | All SV callers | INV | 12 | p13.2 q23.1 | 11873746 | 96556989 | 84683243 | 9 | 18 | 16 | |
| | | BND | 12 | p13.2 | 0 | 11874372 | 11874372 | 6 | 8 | 11 | ETV6::RUNX1 |
| | | | 21 | q22.12 | 34948109 | 46709983 | 11761874 | | | | |
| | | BND | 21 | q22.12 | 34947932 | 46709983 | 11762051 | 19 | 18 | 32 | RUNX1::PRDM7, CBFA2T3 |
| | | | 16 | q24.3 | 90067326 | 90338345 | 271019 | | | | |
| | | BND | 16 | q24.3 | 90067537 | 90338345 | 270808 | 5 | 7 | 8 | PRDM7 |
| | | | 4 | q32.1 | 159785190 | 190214555 | 30429365 | | | | |
| | | BND | 4 | q32.1 | 159785199 | 190214555 | 30429356 | 12 | 17 | 11 | |
| | | | 12 | q23.1 | 96556785 | 133275309 | 36718524 | | | | |
| | | BND | 5 | q23.2 | 128865814 | 181538259 | 52672445 | 6 | 9 | 16 | LRP6::SLC27A6 |
| | | | 12 | p13.2 | 0 | 12123786 | 12123786 | | | | |
| | | BND | 12 | p13.2 | 0 | 11540006 | 11540006 | 13 | 16 | 14 | PHAX::AC007450.2 |
| | | | 5 | q23.3 | 126620210 | 181538259 | 54918049 | | | | |
| | | DEL | 3 | p22.3 p14.2 | 35660443 | 61437807 | 25777364 | 10 | 16 | 11 | ARPP21, FHIT, SETD2 |
| Novel SVs | | DEL | 3 | q26.32 | 177050707 | 177196318 | 145611 | 12 | 5 | 25 | TBL1XR1 |
| | | DEL | 5 | q31.3 | 143197445 | 143402107 | 204662 | 7 | 8 | 19 | NR3C1::ARHGAP26 |
| | | DEL | 6 | q21 | 111426787 | 111563892 | 137105 | 11 | 6 | 19 | TRAF3IP2::REV3L |
| | | DEL | 12 | q21.33 | 91884416 | 92144292 | 259876 | 12 | 8 | 13 | BTG1 |
| | | DEL | 14 | q24.2 | 72752203 | 72892376 | 140173 | 11 | 16 | 15 | DPF3 |
| | | INV | 16 | p12.2 | 21583122 | 22699431 | 1116309 | 13 | 16 | 18 | |
| | | DEL | 18 | q21.1 | 48950471 | 49053159 | 102688 | 6 | 9 | 13 | SMAD7 |
| | | DEL | 18 | q23 | 79384761 | 79516951 | 132190 | 3 | 7 | 10 | NFATC1 |
| | | DEL | 18 | q23 | 75992809 | 76208899 | 216090 | 9 | 5 | 16 | |
| | | DEL | 22 | q11.22 | 22031472 | 22245538 | 214066 | 6 | 9 | 29 | PRAMENP, VPREB1 |
| | | DEL | 22 | q11.22 | 22713200 | 22904992 | 191792 | 5 | 11 | 21 | IGLL5 |
| | ONT and PB | BND | 2 | p11.2 | 89027169 | 89085084 | 57915 | 9 | 13 | 5 | |
| | | | 1 | q21.1 | 146784567 | 146784579 | 12 | | | | |
| | | DUP | 1 | p21.1 | 103561098 | 103676921 | 115823 | 9 | 7 | 3 | |
| | ONT and Illumina | INV | 2 | p11.2 | 88861924 | 90221383 | 1359459 | 7 | 9 | 20 | |
| | Illumina only | DEL | 9 | p21.3 | 20676192 | 23121222 | 2445030 | 20 | 19 | 20 | CDKN2A, FOCAD, IFN, KLHL9 |
| Average read support[a] | | | | | | | | 9.57 | 11.18 | 15.87 | |

[a]Read support = split reads for long-read data and discordant mates for Illumina data.

intron 4; five splicing variants of *RUNX1::PRDM7* were found involving exons 5–11, with the most prevalent fusion transcript taking place between exon 1 of *RUNX1* and exon 9 of *PRDM7* (Fig 4B).

Two out-of-frame fusion genes were found to arise from the balanced t(5;12): *PHAX::AC007450.2* and *LRP6::SLC27A6*; both were detected in both the short- and long-read RNA-seq datasets. In *PHAX::AC007450.2*, which was supported by nine Illumina reads and five IsoSeq reads, exon 4 of *PHAX* was fused with exon 2 of

*AC007450.2*, which lies 85 kb downstream of *ETV6* (Fig 4C). Exon 22 of *LRP6* was fused with exon 2 of *SLC27A6* (Fig 4D). *LRP6::SLC27A6* was supported by one short read and two long reads. However, manual inspection of the reads revealed an additional 11 low-quality split long reads that were discarded by the fusion caller and not reported in the supporting read count.

A fusion transcript arose from the 260-kb deletion del(12)(q21.33q21.33), which resulted in a truncated *BTG1* gene, with a breakpoint in

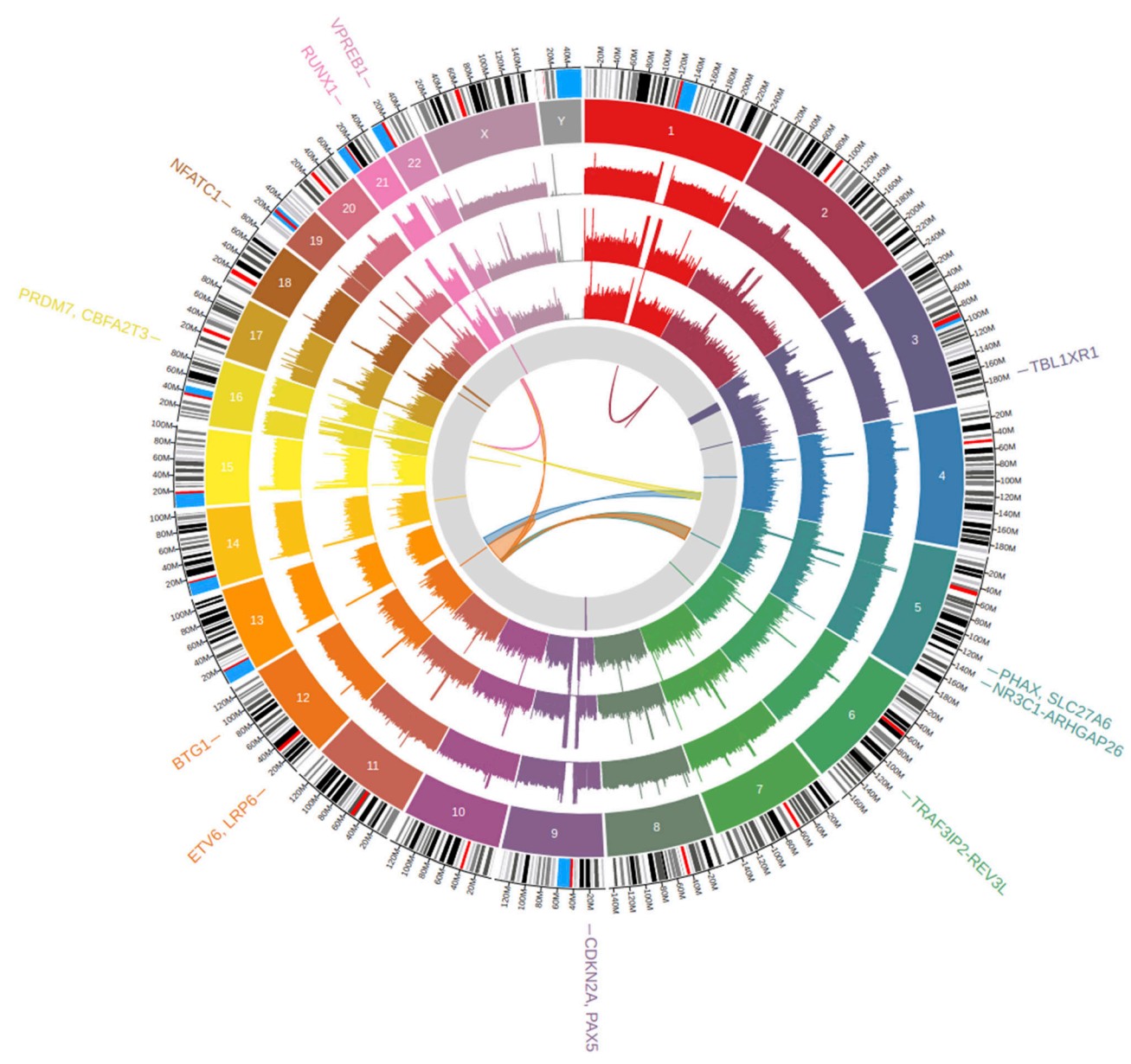

**Figure 3. Large-scale structural variants confirmed in REH.**
The innermost circle shows inversions and interchromosomal translocations. The gray band depicts deletions >100 kb. The rainbow bands show the depth of coverage in PacBio, ONT, and Illumina datasets, respectively, followed by chromosome number. The outermost band indicates the GRCh38 reference cytoband with genes disrupted listed on the outer edge of the plot.

exon 2. In this fusion, the truncated *BTG1* transcript was fused with an expressed non-coding region originating between two long non-coding RNA genes *LINC02404* and *AC090049.1*. The *BTG1::LINC02404/AC090049.1* fusion was only called in the short-read RNA-seq dataset using Arriba, but by no long-read callers. The resulting out-of-frame truncated transcript was highly expressed with 120 supporting Illumina reads and nine IsoSeq reads (Fig 4E).

The 205-kb del(5)(q31.3q31.3) resulted in the antisense transcript *NR3C1::ARHGAP26*, fusing exon 1 of *NR3C1* with exon 20 of *ARHGAP26*. This fusion was only detected by one short-read fusion caller, with

10 supporting Illumina reads, but also found support in the long-read data, with five supporting IsoSeq reads (Fig 4F).

Finally, the 137-kb del(6)(q21q21) resulted in the fusion gene *TRAF3IP2::REV3L*, fusing exon 8 of *TRAF3IP2* to exon 3 of *REV3L*. This fusion was not detected by any long-read callers but was found by three short-read fusion callers and was supported by six Illumina reads and 13 IsoSeq reads (Fig 5G).

Further splicing and read support details can be found in Table S6.

Of note, three of the seven confirmed fusion genes involve the partner genes *ETV6* and *RUNX1* or regions in their immediate vicinity

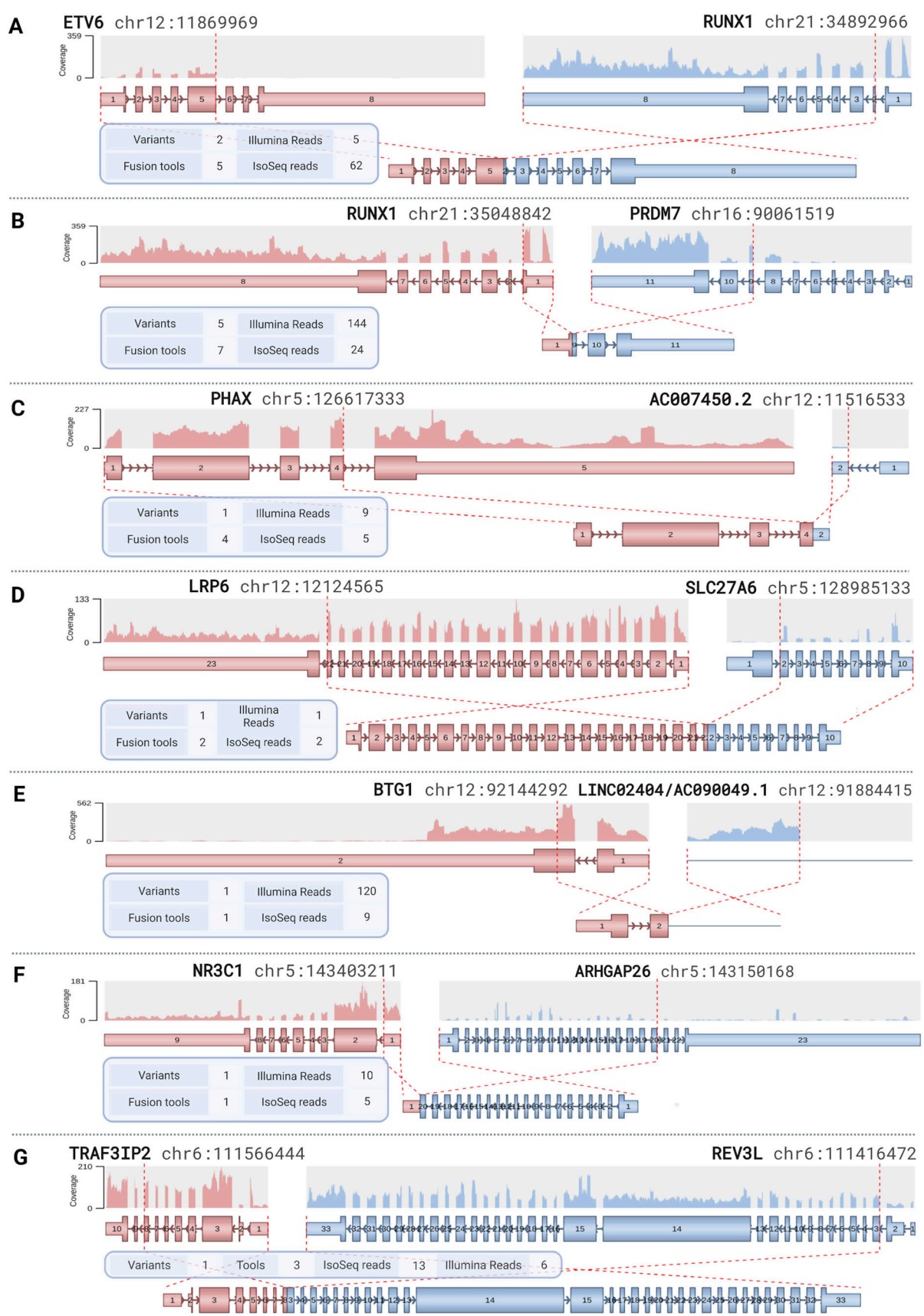

(*ETV6::RUNX1*, *PHAX::AC007450.2*, and *RUNX1::PRDM7*). One homolog of chromosome 21 was found to be involved in both *RUNX1* fusions, whereas both homologs of *ETV6* were involved in different chromosomal aberrations occurring on chromosome 12, with one forming the *ETV6::RUNX1* fusion and the other undergoing a 584-kb deletion between the breaking and fusion events of the t(5;12), deleting the entire gene and leaving no WT *ETV6* in the genome.

The performance of the seven fusion gene calling tools varied widely. The number of fusion gene candidates found by the short-read callers had a wide range: STAR-fusion found only seven, Arriba found 31, and FusionCatcher found 78, while at the higher end of the range were Squid (n = 393) and pizzly (n = 5,520). The two long-read callers returned a large number of candidates: 336 from Cupcake and 4,927 from JAFFAL. The percentage of fusion gene candidates passing automated filtering were as follows: JAFFAL, 0.24%; Squid, 0.25%; pizzly, 0.36%; Cupcake, 1.79%; FusionCatcher, 7.69%; Arriba, 29.03%; and STAR-fusion, 57.14%.

The overall performance of the fusion callers also varied widely, as measured by sensitivity (percentage of the manually confirmed fusion genes detected) and FPR, calculated after discounting fusion gene candidates also found in the GM12878 RNA-seq dataset. Arriba detected each of the seven manually confirmed fusion genes (100.0% sensitivity, but 76.67% FPR), and STAR-fusion had the lowest FPR (50.0%) but only detected three fusion genes (42.86% sensitivity). Squid had the lowest sensitivity rate (14.29%; 1 confirmed gene fusion found), with an FPR of 99.65%. The remaining tools each called three of the confirmed fusion genes (42.86% sensitivity rate) and had an FPR between 94.23% and 99.94% (Table S7).

### Key SNVs conclude the detailed characterization of the REH cell line

Finally, we characterized the SNVs in the REH genome. For SNV discovery, only the Illumina data were used, because of the higher coverage and lower error rate of this dataset in comparison with the long-read datasets (Table S2). Key SNVs, detailed below, were subsequently corroborated in the long-read data.

Mutect2 was run on the Illumina WGS dataset, filtered against a panel of normals from the 1000 Genomes Project and allele frequencies from gnomAD, to produce a set of SNVs that were further filtered to include only variants affected by loss of function or nonsense-mediated mRNA decay (NMD). The resulting set contained 131 variants (Table S8), including a nonsense mutation in exon 5 of GC receptor *NR3C1* (p.Gln528Ter; Fig S11) and a frameshift mutation in exon 8 of *PAX5* (p.A322Rfs*19; Fig S12). Phasing analysis placed the *NR3C1* mutation on the same homolog of chromosome 5 (contig 56) as the t(5;12).

Combining cytogenetics, SV analysis, and SNV detection, our integrative short- and long-read genomic and transcriptomic analysis allows us to provide a detailed characterization of the REH cell line, shown in Fig 5.

## Discussion

Cell lines continue to be a vital resource for functional studies. Several well-established cancer cell lines are highly rearranged, with underreported genomic complexity (Nattestad et al, 2018; Aganezov et al, 2020; Weichenhan et al, 2023). Despite its frequent use for functional and mechanistic studies in leukemia research (Oshima et al, 2020; Diedrich et al, 2021; Cousins et al, 2022; Leo et al, 2022; Wray et al, 2022), a comprehensive analysis of the complex REH genome has never been undertaken. Herein, we used a combination of short- and long-read DNA and RNA sequencing to provide a detailed characterization of the REH cell line. This clarifies the ambiguity left by traditional karyotyping and resolves several known and novel aberrations in the REH genome, which are of potential biological importance.

REH exhibits the *ETV6::RUNX1* fusion gene, which is thought to arise in utero and is alone not sufficient for leukemogenesis (Greaves, 2018). A number of complementary aberrations have been shown to contribute to leukemogenesis, including deletions in the WT copy of *ETV6*; LOH in the p-arm of chromosome 9, coupled with the homozygous deletion of *CDKN2A* and mutations in *PAX5*; and deletions affecting the genes *BTG1* and *TBL1XR1* (Mullighan et al, 2007; Papaemmanuil et al, 2014).

In the present study, both alleles of *ETV6* were affected by aberrations, with one homolog forming the *ETV6::RUNX1* fusion and the other undergoing full deletion proximal to the t(5;12) breakpoint, which forms the *PHAX::AC007450.2* fusion. Of note, *AC007450.2* refers to the Bacterial Artificial Chromosome (BAC) clone RP11-434C1 (https://www.ncbi.nlm.nih.gov/nuccore/AC007450.1), which is 85 kb downstream, and on the opposite strand, of *ETV6* (Tsuzuki et al, 2007; Hoff et al, 2016). *PHAX::AC007450.2* has not been reported in ALL cases previously; however, *PHAX*, a protein-coding gene involved in the nuclear export of small nuclear RNA (Ohno et al, 2000), appears in a single *PHAX-IGH* fusion that was recently reported in a *ETV6::RUNX1*-like case (Li et al, 2021).

LOH on the p-arm of chromosome 9 has been shown to occur in over half of ALL cases, with a third of B-ALL cases exhibiting homozygous deletions of *CDKN2A* (Takeuchi et al, 1997), the latter correlating with poor outcome (Kathiravan et al, 2019). *PAX5*, which encodes BSAP (B-cell–specific activator protein), a transcription factor playing an essential role in B-cell development, has been shown to undergo mutations in over a third of both pediatric and adult B-ALL patients (Familiades et al, 2009). In REH, a frameshift mutation in exon 8 of *PAX5* has been previously observed in multiple studies (Best et al, 2010; Hart et al, 2018) and found to inactivate a critical transactivation domain within BSAP (Dörfler & Busslinger, 1996).

**Figure 4. REH fusion gene breakpoints, visualized by the Arriba module of the nf-core/rnafusion pipeline.**
**(A)** The more highly expressed variant of the two splice variants of *ETV6::RUNX1*, resulting from t(12;21). **(B)** The most highly expressed variant of the five splicing variants of *RUNX1::PRDM7*, resulting from t(16;21). **(C, D, E, F, G)** Fusion gene breakpoints in (C) *PHAX::AC007450.2* and (D) *LRP6::SLC27A6*, both from t(5;12), (E) *BTG1::LINC02404/ AC090049.1* from del(12)(q21.33q21.33), (F) *NR3C1::ARHGAP26*, from del(5)(q31.3q31.3), and (G) *TRAF3IP2::REV3L*, from del(6)(q21q21).

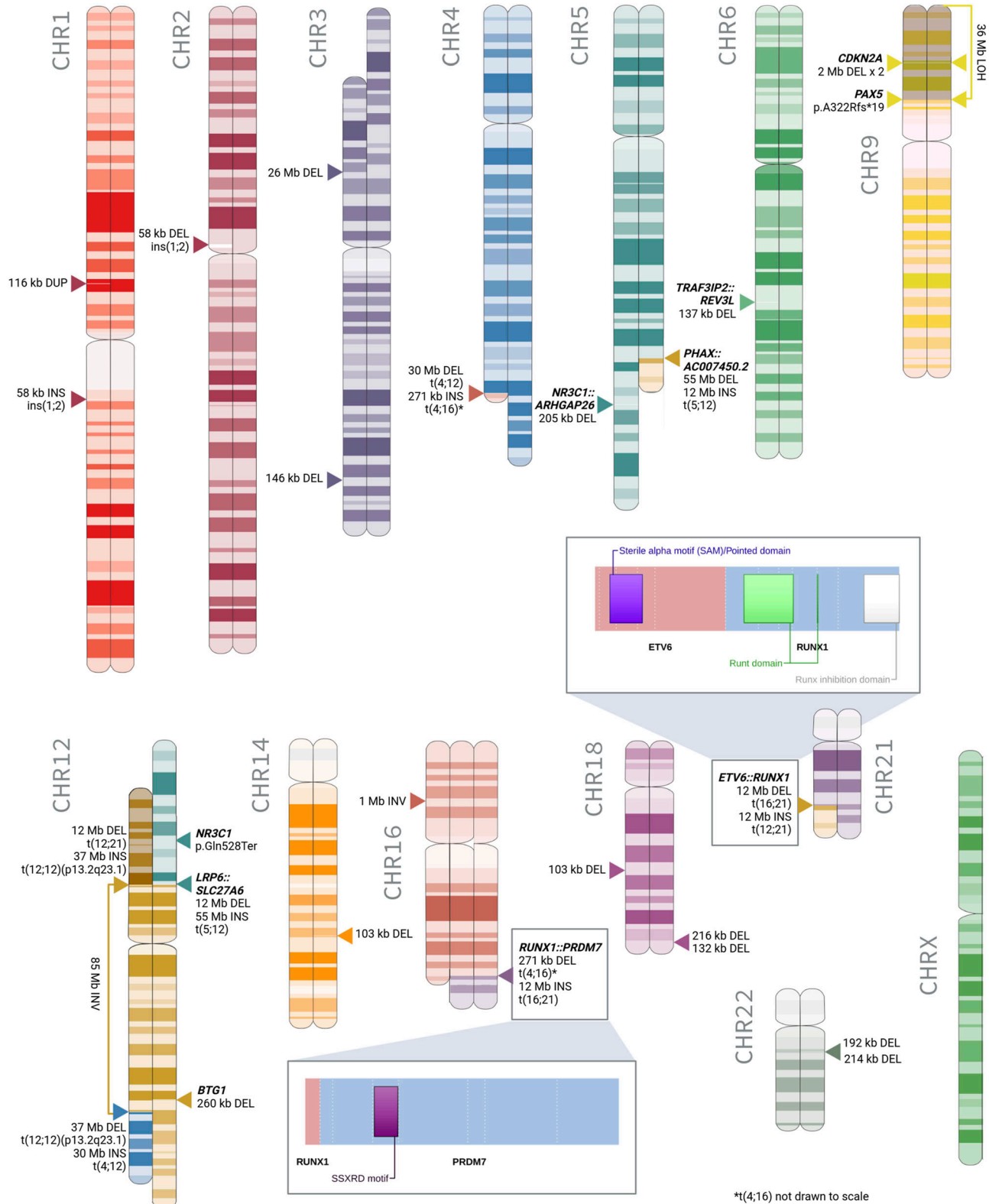

**Figure 5. Aberrant chromosomes of the REH cell line.**
Highlighted are the retained protein domains for ETV6::RUNX1 and RUNX1::PRDM7, resulting from the two REH in-frame fusion genes. Rearrangements that could not be phased to a specific homolog have been rendered on an arbitrary homolog of their respective chromosomes.

The REH cell line comes from a patient at ALL relapse, making it particularly notable that several deletions identified in the present study, affecting genes *TBL1XR1* (146-k deletion on 3q26.32), *NR3C1* (205-kb deletion on 5q31.3), and *BTG1* (260-kb deletion on 12q21.33), have been shown to contribute to its GC drug resistance (van Galen et al, 2010; Jones et al, 2014; Xiao et al, 2019). GCs are a key component of ALL treatment protocols, and the roles of *NR3C1* and *BTG1* expression and perturbation have been widely analyzed in the REH cell line (van Galen et al, 2010; van der Zwet et al, 2021). The *BTG1* deletion identified in the present study, which occurs rarely in ALL patients but frequently co-occurs with *ETV6::RUNX1* (Schwab et al, 2013), has been previously noted in analyses of REH using arrays (Tsuzuki et al, 2007) and PCR (van Galen et al, 2010), whereas *NR3C1* has an established nonsense (p.Gln528Ter) mutation on one allele (Tamai et al, 2022) and a deletion on the other (Grausenburger et al, 2016). Recurrent deletions of *TBL1XR1*, which encodes for nuclear hormone receptor co-repressors, have been shown to contribute to dysregulated gene expression and GC resistance in *ETV6::RUNX1*-positive ALL, including REH (Parker et al, 2008; Jones et al, 2013).

In the present study, we identified a previously unknown REH variant (214-kb deletion on 22q11.22) that deletes the *VPREB1* gene, which codes for part of the surrogate light chain that forms pre-BCRs. *VPREB1* deletions have previously been shown to occur in up to 40% of *ETV6::RUNX1*-positive ALL patients at diagnosis and >70% at relapse (Kuster et al, 2011), contributing to the disordered light chain rearrangement that acts as a leukemogenic trigger (Mangum et al, 2014). The *VPREB1* deletion in REH, which also encompasses part of the pseudogene *PRAMENP*, is adjacent to another 192-kb deletion, affecting the chronic lymphocytic leukemia-associated gene *IGLL5* (Kasar et al, 2015); the gene that lies between these two deletions, *PRAME*, has been identified as a potential therapeutic target for pediatric ALL (Steinbach et al, 2002). Recurrent deletions at the 22q11.22 locus have been previously identified as a prognostic marker of poor outcome for certain B-ALL patients (Mangum et al, 2021), making this a genomic region warranting further research.

Likewise, the long arm of chromosome 18 was the site of recurring deletions. Of the three large-scale deletions found between 18q21 and 18q23, one was found to affect the *NFATC1* gene (132-kb del on 18q23), a member of the *NFAT* family involved in the calcineurin signaling pathway, which has been identified as a potential therapeutic target (Medyouf & Ghysdael, 2008). Meanwhile, the 26-Mb deletion on chromosome 3 encompasses a large number of genes, including *SETD2*, which is associated with chemotherapy resistance and relapse when co-occurring with *ETV6* deletions (Mar et al, 2017; Oshima et al, 2020); *ARPP21*, a frequent deletion in *ETV6::RUNX1*-like ALL cases (Zaliova et al, 2017); and *FHIT*, a tumor suppressor gene that has been shown to be abnormally expressed in ALL (Hallas et al, 1999).

The fusion gene arising from the t(16;21)(q24;q22) is *RUNX1::PRDM7*. This highly expressed fusion occurs on the same *RUNX1* homolog as the canonical *ETV6::RUNX1* fusion. In contrast to previous reports (Lilljebjörn et al, 2014; Ghandi et al, 2019), our data indicate that this fusion is in-frame, retaining the SSXRD motif of PRDM7. Because of the close sequence similarity between *PRDM7* (chromosome 16) and its paralogous copy *PRDM9* (chromosome 5) (Fumasoni et al, 2007), this fusion was misidentified as *RUNX1-*

*PRDM9* by two short-read fusion callers and a mismapping of up to 127 short RNA-seq reads. PRDM9, which shares the SSXRD, SET, and KRAB protein domains with PRDM7, has been implicated in genetic predisposition to pediatric B-ALL (Hussin et al, 2013; Thibault-Sennett et al, 2018), whereas our findings suggest that this association might extend to other members of the *PRDM* gene family.

The fusion callers suggested several other fusion events between *RUNX1* and genes proximal to the genomic breakpoint on 16q24.3. Notably, the *CBFA2T3* gene, 1 Mb downstream of *PRDM7*, was suggested as a fusion partner by both long-read callers. Manual inspection did not confirm the fusions involving *CBFA2T3*; however, there were at least four large indels of 90–800 bp within the gene, as well as a number of smaller indels (<50 bp), suggesting genomic instability in the region.

Aberrations involving chr16q24.3 are frequent in ALL and in acute myeloid leukemia, where one of the well-established subtypes is defined by t(16;21) *RUNX1-CBFA2T3* (Noort et al, 2018). REH has an abnormally high expression of both *CBFA2T3* and *RUNX1* (Rouillard et al, 2016), and the *RUNX1* fusion activates a topological disruption of multiple genes in this region that promotes BCP-ALL proliferation (Jakobczyk et al, 2021).

In assessing the performance of the long- and short-read technologies, we found benefit in integrating a variety of tools and datasets with distinct strengths in detecting variants of different types and sizes (van Belzen et al, 2021). Overall, we found that the ONT dataset was the most useful for SV discovery, with the highest sensitivity and lowest FPR; however, when adjusting for the slightly lower DC, PacBio had a comparable performance. Importantly, both long-read datasets were able to resolve variants in highly repetitive regions that were missed in the short-read data.

We found that SV detection using TIDDIT on Illumina short reads resulted in an FPR of over 98%, limiting its stand-alone utility in SV discovery in the absence of a matched normal control; in particular, a known problem with this method is a high rate of falsely identified translocations (Sedlazeck et al, 2018). However, short paired reads were useful in confirming and resolving breakpoints suggested by the long reads; thus, given the high throughput and low cost, short reads remain a sensible choice for clinical screenings targeting specific variants. Of note, the high FPR and low sensitivity observed in this study are the statistical side effects of using a consensus method combining SV calls from several different methods; the upside to this approach is high precision (Liu et al, 2020).

In assessing fusion detection capabilities, we found short-read RNA-seq data to be more informative than the IsoSeq dataset, largely because of the superior performance of the short-read fusion caller Arriba, which has been shown to outperform both other short-read fusion callers (Creason et al, 2021) and long-read callers in low-coverage contexts (Van Twisk et al, 2022 Preprint).

The present study has several limitations. Firstly, REH has no matched normal sample available, nor are non-leukemic cells present in this cell line, which makes it challenging to differentiate between germline and somatic mutations. Second, the long-read WGS datasets had two different shortcomings. Our long-read datasets had relatively lower coverage (15x PacBio, 18x ONT)

than the Illumina dataset (34x). In addition, the ONT sequencing data were generated using a version R9.4.1 flow cell, resulting in a mismatch rate of 3% and an overall error rate of 6.55%, which is in line with the reported error rates for this flow cell version, but higher than the error rates for the other datasets (<1%). The R10.4 flow cell version and latest updates to chemistry have reduced the average error rate for ONT data to 3.9%, with the potential to both improve SV calling and perform increasingly accurate methylation calling as part of similar integrative analyses (Sigurpalsdottir et al, 2024). Finally, there were challenges with our phasing approach using de novo assembly. Although this approach leveraged the ability of long reads to assemble into multi–megabase-pair contigs, potentially allowing the resolution of chromosomal context for distant SVs, it suffered from misassembly and difficulty resolving haplotypes. It has been previously noted that de novo assembly in an oncological context may struggle to construct accurate contigs because of the complexity and heterozygosity of cancer genomes (Sakamoto et al, 2021). However, where matched tumor–normal samples are available this approach can be highly informative for somatic SV detection (Xiao et al, 2022).

In summary, this study presents a detailed characterization of the REH cell line, cataloging both known aberrations and previously underreported variants and fusion genes. Our findings provide high-quality reference data and highlight the need for comprehensive genomic analysis of commonly used and perturbed cell lines. Furthermore, we have examined the increased informational context of long reads and their enhanced ability to capture large-scale SVs, a promising development for clinical cancer genomics. Finally, we have described multiple aberrations in REH that are frequently observed in high-risk ALL patients, making this cell line a highly relevant model for researchers investigating the biological underpinnings of GC resistance and relapse.

# Materials and Methods

### REH cell line

The cell line REH was obtained from DSMZ (ACC 22) and cultured according to the supplier's specifications. The cell line was verified by Eurofins human STR profiling cell line authentication services. DNA and RNA were extracted from 5 M cells using QIAGEN's AllPrep DNA/RNA Micro Kit. For the long-read WGS libraries, PacBio's Circulomics Nanobind UHMW DNA Extraction was used with an input of 15 M cells.

G-banded karyotyping was performed by the Department of Clinical Genetics, Uppsala University Hospital, Uppsala, Sweden. The cells from the cell culture were harvested following standard cytogenetic methods (KaryoMAX Colcemid Solution in HBSS, Gibco), and chromosome preparations were stained by G-banding trypsin–Giemsa standard procedures (0.25% trypsin, Gibco; Wright's stain, Sigma-Aldrich; Giemsa's stain, VWR Chemicals and Buffer tablets, pH 6.8, Merck). Twenty-five metaphases were analyzed using a Carl Zeiss scanner microscope and Metafer and Ikaros software from MetaSystems. The karyotype was carried out according to the International System for Human Cytogenetic Nomenclature (ISCN) 2020.

### Sequencing

An Illumina PCR-free WGS library was generated with 1,000 ng input DNA and sequenced with PE-150 on a HiSeq X instrument. An Illumina short-read RNA-seq library was generated with 300 ng input total RNA and the TruSeq Stranded Total RNA kit, and sequenced with PE-100 on a NovaSeq 6000 instrument. One ONT WGS library was generated from 40 µg input DNA prepared with the Ultra-Long Sequencing Kit (SQK-ULK001) and sequenced on PromethION 24 using a R9.4.1 flow cell. Two PacBio IsoSeq RNA-seq datasets and one PacBio SMRTbell HiFi WGS dataset were prepared with SMRTbell Express Template Prep Kit 2.0, Sequel II Binding Kit 2.0, and Sequel II Sequencing Plate 2.0. For the IsoSeq library, 300 ng input RNA was used for each of the two libraries; a varying bead ratio was used to generate one library with standard-length transcripts and one library with full-length transcripts. Full-length non-chimeric (FLNC) reads were generated using PacBio tools ccs and lima. For the PacBio WGS, 10 µg input DNA was used to create a continuous long-read (CLR) library and 15 µg to create a HiFi library; the resulting CCS reads from both libraries were merged for analysis.

One RNA-seq dataset was also generated from GM12878 (B lymphoblastoid cell line) by merging four Illumina short-read RNA-seq libraries, each generated using 500 ng of input total RNA with the TruSeq Stranded Total RNA with Ribo-Zero depletion, and sequenced on a NovaSeq 6000 instrument.

### Data analysis

Reads were aligned to the reference genome version GRCh38, followed by analysis with tools for SNV, SV, and fusion gene detection (Table 1). For gene annotation, GENCODE V37 (Ensembl 103) was used. Sequencing statistics (Table S2) were generated using Alfred. Versions for all software packages are listed in Table S9.

For visualization, we ran Copycat on the short- and long-read BAM files to bin the read coverage in preparation for visualization. Visualization was performed with SplitThreader (Nattestad et al, 2016 Preprint), Ribbon (Nattestad et al, 2021), and IGV. We used SplitThreader to visualize interchromosomal SVs across the Illumina, PacBio, and ONT callsets, whereas Ribbon was used for inspection of intrachromosomal features and split-read analysis. Circos plots were drawn using Circa. Heatmaps were generated with the chromoMap R package.

### WGS data analysis

The short-read WGS data were mapped and processed with the nf-core/sarek pipeline (Garcia et al, 2020). PacBio CCS reads were mapped and sorted using pbmm2. After adapter sequences were trimmed using Porechop, the ONT reads were aligned using minimap2.

For the variant allele frequency plots (Figs 1 and S2), three sets of SNV calls were generated by running DeepVariant on the WGS datasets (in conjunction with PEPPER, for ONT).

For the filtered SNV callset (Table S8), Mutect2 was run on the Illumina alignments in tumor-only mode, filtered against a panel of normals generated from the 1000 Genomes Project and allele

frequencies from gnomAD, both provided by the GATK Best Practices bundle. Calls with a population frequency of over 0.0001 were excluded. SNV calls were subsequently annotated with SnpEff and filtered for loss of function or NMD affecting at least 25% of the gene's transcripts.

TIDDIT (Eisfeldt et al, 2017) was used for SV detection in the short-read data. In both long-read WGS datasets, Sniffles2 (Sedlazeck et al, 2018) was run with non-germline mode enabled. SURVIVOR was used to filter and merge VCF files using a minimum size of 100 bp and excluding variants found in the ENCODE Blacklist. For consensus callsets, SVs were required to agree on type and strand, with breakpoints at a maximum allowed distance of 1 kb.

For the evaluation of large-scale variants and translocations, the original callsets were pre-filtered, retaining SV candidates with either length > 100,000, or of a breakend type and mapping to two different chromosomes; discarded candidates were excluded from accuracy statistics. The remaining candidates were further filtered, requiring them to meet all the following criteria: (1) passing all quality filters set by SV calling software (2) a minimum of five supporting reads, with the number of supporting reads >20% of the average depth of coverage (DC) for the dataset and (3) position coverage no greater than 150% of the average DC for the dataset.

Of the remaining candidates, the following were selected for manual inspection: (1) all candidates with support in at least one Sniffles callset and (2) TIDDIT candidates with exactly 15 supporting reads, providing a pseudorandom sampling from this callset. SV candidates were confirmed visually in IGV, requiring the presence of split long reads and/or discordant mates in at least two of the three SV callsets for confirmation. Final candidates were screened against gnomAD SVs v2.1 to exclude possible germline variants.

The de novo assembly used for SV phasing was created with Flye and Medaka on ONT ultralong reads. Adapter sequences were trimmed from the ONT data before assembly using Porechop. We in addition trimmed the first 50 bp from all reads and removed all reads shorter than 500 bp using NanoFilt (De Coster et al, 2018). After assembly using Flye, polishing was performed with Medaka, which uses neural networks and is trained on specific configurations of ONT pore type, instrument, and basecaller. This assembly was polished using the Medaka model r941_prom_hac_g507. The quality of the assemblies was assessed with QUAST (Mikheenko et al, 2018). The assembly was phased using Hapdup, then realigned to the reference genome, and subjected to SV calling using Hapdiff.

### RNA-seq analysis

The nf-core/rnafusion pipeline was run on short-read REH RNA-seq data to call and visualize putative fusion genes. This pipeline uses five different fusion tools: Arriba, FusionCatcher, pizzly, Squid, and STAR-fusion. The nf-core/rnafusion pipeline was also run on the GM12878 short-read RNA-seq dataset to generate a set of non-leukemic fusion gene candidates for filtering.

Putative fusion genes were called in the long-read IsoSeq data with two different workflows. First, cDNA_Cupcake was used in conjunction with clustering using isoseq3 and fusion gene classification using SQANTI3. Second, the JAFFAL pipeline was run with default settings.

Candidate fusion genes were programmatically pre-filtered as follows. All candidate fusion events called by rnafusion in the GM12878 cell line were removed and excluded from accuracy statistics. The remaining candidate fusion genes were required to have different partner genes and to fulfill any of the following criteria: (1) containing one or more known ALL-associated genes (Marincevic-Zuniga et al, 2017), and supported by at least five reads (2) called by at least one short-read fusion detection tool and one long-read fusion detection tool and (3) called by at least three short-read fusion detection tools, or (4) supported by at least 10 long reads. The remaining candidates were inspected manually in IGV.

## Data Availability

The sequencing datasets and BAM files generated and/or analyzed during the current study are available in NCBI's Sequence Read Archive (SRA) under BioProject accessions PRJNA600820 and PRJNA834955 (Table S10). Supplementary data are available at Lysenkova Wiklander et al (2024). Analysis scripts in Bash, Python, and R are available at Lysenkova Wiklander (2024).

## Supplementary Information

## Acknowledgements

This project was funded by the Swedish Research Council (#2019-01976), the Swedish Childhood Cancer Fund (#PR2019-0046, #HFT2023-0011), and the Göran Gustafsson Foundation. Sequencing was performed at the National Genomics Infrastructure (NGI) at SciLifeLab in Uppsala. NGI is funded by SciLifeLab, the Swedish Research Council RFI, and the Knut and Alice Wallenberg Foundation. The computations were enabled by resources provided by the National Academic Infrastructure for Supercomputing in Sweden (NAISS) and the Swedish National Infrastructure for Computing (SNIC) at the Uppsala Multidisciplinary Center for Advanced Computational Science (UPPMAX) partially funded by the Swedish Research Council through grant agreements #2022-06725 and #2018-05973. The authors would like to acknowledge Elin Övernäs and Johanna Lagensjö for their assistance with short-read sequencing, Sara Ekberg, Pontus Larsson, Rikard Erlandsson, and Susanne Reinsbach for bioinformatics support, and Mai-Britt Mosbech, Susana Häggkvist, and Anna Petri for assistance with high molecular weight DNA extraction and long-read sequencing. The graphical abstract was created with BioRender.com and is licensed under CC-BY-NC-ND.

### Author Contributions

M Lysenkova Wiklander: conceptualization, data curation, software, formal analysis, investigation, visualization, project administration, and writing—original draft.
G Arvidsson: data curation, methodology, and writing—review and editing.
I Bunikis: data curation, formal analysis, and writing—review and editing.

A Lundmark: data curation, formal analysis, and visualization.

A Raine: conceptualization, resources, and methodology.

Y Marincevic-Zuniga: conceptualization, resources, formal analysis, and methodology.

H Gezelius: data curation and validation.

A Bremer: formal analysis, visualization, and writing—review and editing.

L Feuk: conceptualization, funding acquisition, investigation, methodology, project administration, and writing—review and editing.

A Ameur: conceptualization, data curation, formal analysis, investigation, methodology, and writing—review and editing.

J Nordlund: conceptualization, funding acquisition, supervision, investigation, methodology, project administration, and writing—original draft.

## Conflict of Interest Statement

The authors declare that they have no conflict of interest.

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
