## [Reviewer comments · Life Science Alliance]

Life Science Alliance

A multiomic characterization of the leukemia cell line REH using short- and long-read sequencing

Mariya Lysenkova Wiklander, Gustav Arvidsson, Ignas Bunikis, Anders Lundmark, Amanda Raine, Yanara Marincevic-Zuniga, Henrik Gezelius, Anna Bremer, Lars Feuk, Adam Ameer, and Jessica Nordlund

DOI: <https://doi.org/10.26508/lsa.202302481>

Corresponding author(s): *Jessica Nordlund, Uppsala University*

Review Timeline:	Submission Date:	2023-11-14
	Editorial Decision:	2024-01-08
	Revision Received:	2024-04-10
	Editorial Decision:	2024-04-25
	Revision Received:	2024-05-02
	Accepted:	2024-05-02

Transaction Report:

January 8, 2024

Re: Life Science Alliance manuscript #LSA-2023-02481

Jessica Nordlund
Uppsala University
Sweden

Dear Dr. Nordlund,

Thank you for submitting your manuscript entitled "A complete digital karyotype of the B-cell leukemia REH cell line resolved by long-read sequencing" to Life Science Alliance. The manuscript was assessed by expert reviewers, whose comments are appended to this letter. We invite you to submit a revised manuscript addressing the Reviewer comments.

Thank you for this interesting contribution to Life Science Alliance. We are looking forward to receiving your revised manuscript.

Sincerely,

B. MANUSCRIPT ORGANIZATION AND FORMATTING:

Reviewer #1 (Comments to the Authors (Required)):

Wiklander and colleagues submit "A complete digital karyotype of the B-cell leukemia REH cell line resolved by long-read sequencing" for consideration in the journal Life Science Alliance. The authors report a multi-omics structural variation characterization of the leukemia cell line REH. Cell lines that are used extensively for research and for modeling disease benefit from having the most complete characterization possible and, as such, this manuscript adds to the literature in its detailed analysis of this cell line.

The manuscript details a combined approach using genome sequencing (short and long read) plus transcriptomics to further characterize SVs detected in this cell line and RNA evidence to validate the functional consequences of various SVs. The data illustrate the considerable complexity detected using a suite of approaches in addition to the high false positive rates that would be present using only a single technique in a de novo setting. As such, the combination of the techniques leads to validated high resolution analysis of this cell line.

While the technical and illustrative components of this manuscript are sound, the text and cytogenetic interpretations need improvement. Please find below a list of general and nomenclature improvements that are necessary for publication of this manuscript in LSA.

General Comments.

Title

"A complete digital karyotype of the B-cell leukemia REH cell line resolved by long-read sequencing"

This reviewer would recommend considering changing the title of this manuscript. The term 'digital karyotype', in this reviewer's opinion, is problematic and has been used/misused with several techniques including for chromosomal microarray analysis - where its use was completely inaccurate. The authors acknowledge in the manuscript that several regions of the genome are not analyzed (e.g. ENCODE Black List) and I also assume that centromeric and constitutive heterochromatic regions were likely also not analyzed. It might be most accurate to suggest "A structural genomic and transcriptomic characterization of the B-cell leukemia REH cell line using long and short read sequencing".

bp, kbp and Mbp

According to the NIH the approved abbreviations for kilobase and megabase are kb and Mb. Using bp is appropriate when talking about base pairs. E.g. 20 bp. However 20 kilobases would be 20 kb. Suggest abbreviations be adjusted.

Page 3

Line 4 "complex four-way translocation t(4;12;21;16)"

The term complex is not correctly used here. As the authors detail in the results the 4 segment rearrangement occurs from a double strand break on each of the 4 chromosomes followed by re-joining. This is essentially a balanced 4-way reciprocal rearrangement. By the ISCN, related, multiple segment events are considered as "one" abnormality and do not meet the threshold for complexity (alone). In order to use the term "complex" this would suggest that a chromoanasythetic event has taken place. If the rearrangement is simply a 4-way rearrangement I would suggest saying simply "a four-way rearrangement" or a "a four segment translocation" as the introduction of the term complex implies other things. The authors note that there is some additional complexity around this rearrangement including the inv(12) and del(12q) and the additional copy of the der(16). These abnormalities add to the overall complexity of the karyotype but these abnormalities may have arisen subsequently and not represent the original stemline. Since the issue of a "complex karyotype" is generally used in the context of a karyotype with multiple abnormalities I find it imprecise to use the term complex to also describe rearrangements.

2nd paragraph "Traditional karyotyping and banding techniques can resolve structural variants to a resolution of 5Mbp".

5 Mb is an optimistic lower limit of resolution for high resolution banding (>600 bands). For most cancer karyotyping (300-400 bands) the lower limit of resolution is probably between 10 Mb and 30 Mb. For example, your Figure 1 shows a G-banded karyotype with 350-400 band resolution. A lot depends on the chromosome quality (banding and morphology) and if sufficient metaphases with high enough resolution are present. Not to mention depending on the type of rearrangement and the area of the chromosome (light or dark region, subtelomere etc) resolution can be quite variable. Therefore, I think this statement should

be tempered to suggest that karyotyping can achieve a lower limit of resolution "down to" 5 Mb or in cancer analysis the lower limit of resolution is between 5-30 Mb.

"Whole genome and transcriptome sequencing (WGTS) have become increasingly important as a comprehensive clinical test for accurately resolving genomic breakpoints of structural variants (SVs) and fusion genes in cancer genomes, providing significant improvements in diagnostic yield over analog methods 13-15."

This statement is problematic. While WG and WGTS are increasingly being used in the clinical space there is no widespread consensus that either technique can be used as a "comprehensive clinical test" for structural variants. Your own manuscript, which uses 3 sequencing technologies has a relatively low concordance on SVs between the methods. WG and WGTS applications are more utilized for SNV and copy number detection and for fusion detection but are not used routinely for SV detection. You could suggest that WG and WGTS are increasingly accessible tools for evaluating SVs in high resolution beyond the capabilities of current conventional testing (e.g. karyotype, FISH) but to suggest it is 'clinical-grade' is inaccurate.

"Sequencing-based diagnostic approaches are predominantly based on short-read next generation sequencing (NGS) technologies, which have benefits in low cost, high accuracy, and extensively validated computational pipelines 16-18." I believe this statement is conflating multiple points. Short read Genome Sequencing (srGS) approaches for SNV detection are widely used clinically and have validated computational pipelines. srGS pipelines for SV detection ARE NOT widely validated or used clinically. Low cost is also a somewhat "relative" term. Generally accepted depth for genome sequencing 30-80x is really insufficient for somatic applications with LLOD for VAFs in the 15-25% range for SVs and likely even lower for copy number estimation. I think the point that should be made is that while srGS can be used for SV detection and has lower costs and traditionally had higher accuracy than lrGS suffers from the limitations set forth in the sentence that follows in the manuscript.

Page 4

"to produce a complete digital karyotype"
same point as before. Consider another terminology.

Page 5

The following abbreviations are not defined:
FLNC, CLR and CCS.

Page 9

"We used G-banding to confirm..."
G-banding is a method for staining chromosomes. Not an SV analysis technique. G-banded karyotyping is the analysis technique.

In the supplemental materials referred to on page 9 - Supplemental Table S2 this reviewer notes several errors in the ISCN.

From S2

Authors Version

46, X, -X, dup(1)(p21.1), t(1;2)(q21.1;p11.2)-inv(2)(p11.2), del(3)(p14.2p22.3), del(3)(q26.1), del(3)(q26.32), t(4;12;21;16)(q32.1;p13.2;q22.12;q24.3)-inv(12)(p13.2q23.1), del(5)(q31.3), t(5;12)(q23.2-q23.3;p13.2), del(6)(q21), del(12)(q21.33), del(14)(q24.2), inv(16)(p12.2), der(16)t(16;21)(q24.3;q22.12) x 2, del(18)(q21.1), del(18)(q23) x 2, del(22)(q11.22) x 2

Reviewers Suggestion

46,X,-X,dup(1)(p21.1p21.1),der(1)t(1;2)(q21.1;p11.2),der(2)t(1;2)inv(2)(p11.2p.11.2),del(3)(22.3p14.2),del(3)(q26.1),del(3)(q26.32q26.32),der(4)t(4;12;21;16)(q32.1;p13.2;q22.12;q24.3),del(5)(q31.3q31.3),t(5;12)(q23.2-q23.3;p13.2),del(6)(q21q21),der(12)t(4;12;21;16)inv(12)(p13.2q23.1)del(12)(q21.33),del(14)(q24.2q24.2),inv(16)(p12.2p12.2),der(16)t(4;12;21;16)x2,del(18)(q21.1q21.1),del(18)(q23q23)x2,del(22)(q11.22q11.22)x2

Detailed list of changes to ISCN with justification.

1. all spaces removed.
2. dup(1)(p21.1p21.1) is correct for a small segmental duplication.
3. t(1;2)(q21.1;p11.2)-inv(2)(p11.2). This is not a defined nomenclature style in the ISCN. der(1)t(1;2)(q21.1;p11.2),der(2)t(1;2)inv(2)(p11.2p.11.2) is a better possibility. If the inv(2)(p11.2p11.2) is closer to the p-terminal than the t(1;2) breakpoint it should be listed first, e.g. der(2)inv(2)(p11.2p.11.2)t(1;2). This nomenclature is correct only for a balanced translocation. If translocation is unbalanced, as suggested by authors, needs to be clarified what are the gains or losses to correctly write nomenclature. I could not ascertain this from the data presented. N.B. detailed breakpoints on the t(1;2) not required as they were defined on the der(1). **see additional points regarding page 12 on this rearrangement - need to be clarified 1) exact mechanism 2) fig 5 3) ISCN
4. del(3)(p14.2p22.3) should be del(3)(p22.3q14.2) breakpoints are always from pter to qter in the ISCN.
5. del(3)(q26.1) - should be del(3)(q26.1q26.1) per fig 5.
6. Same point above for del(3)(q26.32)
7. For the t(4;12;21;16) since the 12 has a compound abnormality it cannot be described without using der. I have expanded the

nomenclature above.

8. By karyotype (figure) the del(5)(q31.3) does not appear to be a terminal deletion as that would be visible by karyotype. I have corrected the ISCN to read as a segmental deletion on chromosome 5 as I am assuming this is correct (also per Fig 5). Also, when both chromosome homologues are involved in rearrangements it is often beneficial to differentiate one from the other by underlining the chromosome number when alternate homologues are involved in different rearrangements. For example, the t(5;12) involves one homologue, but the del(5q) is on the other and so the "5" can be underlined for clarity. Apologies for not showing this however the text editing capabilities of the review platform do not allow. Please see any relatively recent version of the ISCN for detailed instructions - underlining (single) Used to distinguish homologous chromosomes (ISCN sections 4.1, 9.2.3, 9.2.17.1. <https://iscn.karger.com/Home/Overview> N.B. while ISCN 2020 requires a subscription previous version can be downloaded as PDFs.

9. del(6)(q21) changed to del(6)(q21q21) to represent a segmental deletion as opposed to a terminal deletion.

10. By fig 5 the del(12q) a part of the der(12) with the t(4;12;21;16). It can be added to the der(12) nomenclature der(12)t(4;12;21;16)inv(12)(p13.2q23.1)del(12)(q21.33).

11. del(14)(q24.2) changed to del(14)(q24.2q24.2) assumed, as above, segmental as loss of 1/3 for 14q would be visible.

12. inv(16)(p12.2) changed to inv(16)(p12.2p12.2) to indicate an inversion within the same band. May wish to underline this if known on other homologue.

13. der(16)t(16;21)(q24.3;q22.12)x2 changed to der(16)t(4;12;21;16)x2 to indicate to copies of the der(16).

14. del(18)'s and del(22) nomenclature corrected by comparison with Figure 1 and Figure 5.

Page 10

"a 26mbp del(3)(p22.3;p14.2)"

should read "26 Mb del(3)(p22.3p14.2)

"a complex four-way rearrangement between chromosomes 4, 12, 21, and 16, resulting in the ETV6-RUNX1 fusion gene"

refer to my previous point about complex. Also ETV6-RUNX1 should be ETV6::RUNX1.

"involve two different alleles of chromosome 12"

I would suggest using the term "homologue" instead of allele. Allele refers to two version of gene. In this case you are saying that the t(5;12) and the t(4;12;21;16) involve both chromosome 12 homologues.

"as the two different p-arms of chromosome 12 (from p13.2 to the telomere)" - would suggest p-terminal instead of telomere.

"The p-arm of chromosome 12 (12Mbp), with breakpoint at p13.2 in intron 5 of ETV6, was translocated to chr21q22.12, resulting in the canonical, subtype-defining fusion gene ETV6-RUNX1." - I would suggest

"The p-arm of chromosome 12, with breakpoint at p13.2 in intron 5 of ETV6, was translocated to chr 21q22.12, resulting in the canonical, subtype-defining fusion gene ETV6::RUNX1."

Page 11

"completing the circular exchange." - suggest 'completing the reciprocal exachange'.

"While this evidence supports the presence of two copies of der(16)t(16;21)" - with the updated ISCN you can now refer to this as simply the "der(16)" and not use the incorreect nomenclature t(16;21).

"were previously unidentified in the analog karyotypes" - the term "analog" seems out of place. I would remove all instances of the term 'analog karoytype(s)' in favour of simply 'karoytype(s)'.

Page 12

"We identified a pair of rearrangements on chromosome 2 including a 1.4 Mbp inversion at p11.2 and an adjacent, unbalanced 58 kb translocation resulting in derived chromosome der(1)t(1;2)(q21.1;p11.2)."

An unbalanced translocation would be associated with a huge region of copy number imbalance. If there is no detected copy number imbalance is this because the t(1;2) is at low VAF or is it possibly an insertion?

Figure 5 does not show a translocation t(1;2). Also, the karyotype in Figure 1 does not show a t(1;2). Is this translocation a sideline? Or with the presenece of a deletion is it possible that it is in fact an insertion? Some comment needs to be made regarding why the t(1;2) is not shown on Fig 5 or the figure corrected.

Page 14

Comprehensive karyotype - please update the nomenclature in this section.

Reviewer #2 (Comments to the Authors (Required)):

This manuscript describes the generation of a genome and transcriptome catalogue of a B-cell leukemia (ALL) cell line, REH. This cell line has been used for various in vitro experimental purposes as a disease model for a long time. However, its precise genomic karyotypes still remaining elusive. In this paper, the authors firstly conducted a long read (PacBio and ONT) and short read (Illumina) whole genome sequencing of this cell line. The authors identified previously suspected and novel breakpoints of structural variations (SVs). Those SVs include the key genes of ALL, such as BTG1 and TBL1XR as well as NR3C1. Possibly relevant SVs in the genes of VPREB1 and NFATC1 were also identified. To further characterize whether any of these identified SVs should be transcribed as a fusion gene transcript, the authors conducted a long read RNA sequencing analysis (Iso seq of PacBio). They successfully identified several fusion gene transcripts and some formed a fusion transcript occurring at the in-frame position.

Overall, I full appreciate that the importance of this data resource should be important to the researchers using this particular cell line. However, I also wonder to what extent this paper should be useful among general readers. Also, I have to point out that the analysis as presented here should not be deep sufficiently deep to render this dataset "comprehensive" or "conclusive".

Major points:

1. I'm afraid that the detected SVs are not "comprehensive". Not all the SNVs residing in this cell line is not represented. Or at least, the precise evaluation on the precision and recall rates are not validated, perhaps using other well-characterized material. More practically, I do not think it is always good idea to pick up the SVs which were detected from PacBio/ONT and Illumina in a (three-way) overlapping manner. For example, the largest advantage in using the long reads should lie in the cases which the short reads have overlooked, but this population would be missed when the overlapping population was picked. However, on the other hand, the data would contain substantial errors if all the results solely based on the long reads were picked up without any filtration. Therefore, a more careful filtration strategy should be considered.

2. Statistics of the sequencings are not detailed. Especially, the read length and the sequencing fidelity of ONT are important. It is potentially advantageous that the ultra-long reads of ONT were employed in this study, similar to the T2T project. However, it is not clear how this advantages are utilized.

3. In addition to SVs, SNVs should be also characterized (or validated if this part is previously characterized, perhaps in CCLE). Also, as the normal counterpart of this cell line is missing, at least, some efforts should be made to exclude its germline variations, both for SVs and SNVs.

Minor points:

4. For identifying aneuploidy regions, particularly careful evaluation should be needed. The loss of whole X chromosome may be the easiest case. However, a wide variety of other factors, which may affect in combination, can be assumed (for example, duplication + loss= seemingly diploid is a very common event). For this purpose, the ration of heterozygous SNPs should give a useful information.

5. Similarly, chromosomal context of each of the SVs can be resolved by further inspecting their surrounding SNPs, taking advantages of the long read (by "phasing" analysis).

6. For the possible "fusion" transcripts, do they likely to have any functional relevance, judging from the transcriptome data, which also represents the gene expression pattern?

7. I wonder if there is any sign for possible emergence of a minor sub-population? The cell line may not be stable after long history of cell culture.

8. I'd like to suggest that DNA methylation call is also possible, which may render this dataset represent more multi-omics features.

Re. manuscript #LSA-2023-02481

Response to Reviewers

Dear Editor,

We would like to thank you and the reviewers for the valuable feedback on our manuscript, now titled "*A multiomic characterization of the leukemia cell line REH using short- and long-read sequencing*". We have implemented most of the suggested changes, taking into consideration compliance with the journal guidelines. We address the changes point-by-point below in blue. We appreciate the opportunity to submit our revisions to the *Life Science Alliance*.

Comments from Reviewer #1

1. *Title "A complete digital karyotype of the B-cell leukemia REH cell line resolved by long-read sequencing" This reviewer would recommend considering changing the title of this manuscript. The term 'digital karyotype', in this reviewer's opinion, is problematic and has been used misused with several techniques including for chromosomal microarray analysis - where its use was completely inaccurate. The authors acknowledge in the manuscript that several regions of the genome are not analyzed (e.g. ENCODE Black List) and I also assume that centromeric and constitutive heterochromatic regions were likely also not analyzed. It might be most accurate to suggest "A structural genomic and transcriptomic characterization of the B-cell leukemia REH cell line using long and short read sequencing".*

Due to the 100-character limit specified in the author guidelines, we changed the title to a shorter version of the reviewer's suggestion: "A multiomic characterization of the leukemia cell line REH using short- and long-read sequencing". The longer summary blurb further reflects the reviewer suggestion: "Long and short-read sequencing are integrated to detail the genomic and transcriptomic characteristics of the cell line REH, a model for relapsed pediatric B-cell acute lymphoblastic leukemia."

Elsewhere in the manuscript, "digital karyotype" has been replaced with "digital characterization", and the words "complete" or "comprehensive" have been replaced with "detailed" or "thorough".

2. *bp, kbp and Mbp. According to the NIH the approved abbreviations for kilobase and megabase are kb and Mb. Using bp is appropriate when talking about base pairs. E.g. 20 bp. However 20 kilobases would be 20 kb. Suggest abbreviations be adjusted.*

All instances of "kbp" and "Mbp" in the main manuscript, supplementary and figures have been changed to "kb" and "Mb", respectively, and now include a preceding space.

3. *Page 3 Line 4 "complex four-way translocation t(4;12;21;16)" The term complex is not correctly used here. As the authors detail in the results the 4 segment rearrangement*

occurs from a double strand break on each of the 4 chromosomes followed by re-joining. This is essentially a balanced 4-way reciprocal rearrangement. By the ISCN, related, multiple segment events are considered as "one" abnormality and do not meet the threshold for complexity (alone). In order to use the term "complex" this would suggest that a chromoanasyntetic event has taken place. If the rearrangement is simply a 4-way rearrangement I would suggest saying simply "a four-way rearrangement" or a "a four segment translocation" as the introduction of the term complex implies other things. The authors note that there is some additional complexity around this rearrangement including the inv(12) and del(12q) and the additional copy of the der(16). These abnormalities add to the overall complexity of the karyotype but these abnormalities may have arisen subsequently and not represent the original stemline. Since the issue of a "complex karyotype" is generally used in the context of a karyotype with multiple abnormalities I find it imprecise to use the term complex to also describe rearrangements.

We agree that this usage of the word "complex" is imprecise. We have removed its use throughout the manuscript when describing rearrangements, and have limited its use to only describing the genome as a whole.

- 4. 2nd paragraph "Traditional karyotyping and banding techniques can resolve structural variants to a resolution of 5Mbp". 5 Mb is an optimistic lower limit of resolution for high resolution banding (>600 bands). For most cancer karyotyping (300-400 bands) the lower limit of resolution is probably between 10 Mb and 30 Mb. For example, your Figure 1 shows a G-banded karyotype with 350-400 band resolution. A lot depends on the chromosome quality (banding and morphology) and if sufficient metaphases with high enough resolution are present. Not to mention depending on the type of rearrangement and the area of the chromosome (light or dark region, subtelomere etc) resolution can be quite variable. Therefore, I think this statement should be tempered to suggest that karyotyping can achieve a lower limit of resolution "down to" 5 Mb or in cancer analysis the lower limit of resolution is between 5-30 Mb.*

We have updated the language and references accordingly: "Traditional karyotyping and banding techniques struggle to resolve structural variants under 5-10 Mb in size, while the resolution of FISH and microarray analyses can go down to approximately 150 kb (Rassekh et al, 2008; Dwivedi et al, 2014)."

- 5. "Whole genome and transcriptome sequencing (WGTS) have become increasingly important as a comprehensive clinical test for accurately resolving genomic breakpoints of structural variants (SVs) and fusion genes in cancer genomes, providing significant improvements in diagnostic yield over analog methods 13-15." This statement is problematic. While WG and WGTS are increasingly being used in the clinical space there is no widespread consensus that either technique can be used as a "comprehensive clinical test" for structural variants. Your own manuscript, which uses 3 sequencing technologies has an relatively low concordance on SVs between the*

methods. WG and WGTS applications are more utilized for SNV and copy number detection and for fusion detection but are not used routinely for SV detection. You could suggest that WG and WGTS are increasingly accessible tools for evaluating SVs in high resolution beyond the capabilities of current conventional testing (e.g. karyotype, FISH) but to suggest it is 'clinical-grade' is inaccurate.

It is true that the clinical use of WGTS for SV and fusion gene detection is in its infancy. We have revised this statement to clarify this point: "Combined whole genome and transcriptome sequencing (WGTS), capable of more accurately resolving clinically relevant structural variants (SVs) and fusion genes than conventional techniques, is being increasingly explored as a diagnostic tool to detect such aberrations in a clinical setting".

- 6. "Sequencing-based diagnostic approaches are predominantly based on short-read next generation sequencing (NGS) technologies, which have benefits in low cost, high accuracy, and extensively validated computational pipelines 16-18." I believe this statement is conflating multiple points. Short read Genome Sequencing (srGS) approaches for SNV detection are widely used clinically and have validated computational pipelines. srGS pipelines for SV detection ARE NOT widely validated or used clinically. Low cost is also a somewhat "relative" term. Generally accepted depth for genome sequencing 30-80x is really insufficient for somatic applications with LLOD for VAFs in the 15-25% range for SVs and likely even lower for copy number estimation. I think the point that should be made is that while srGS can be used for SV detection and has lower costs and traditionally had higher accuracy than lrGS suffers from the limitations set forth in the sentence that follows in the manuscript.*

We agree that these benefits of short-read NGS are relative specifically to long-read sequencing. We have revised this statement accordingly, once again emphasizing that the clinical use of sequencing for SV detection is in an exploratory phase: "Until recently, clinical variant detection using WGTS has been predominantly assessed using short-read next generation sequencing (NGS) due to its lower cost, higher accuracy, and more extensively validated computational pipelines compared to long-read technologies".

- 7. Page 4 "to produce a complete digital karyotype" same point as before. Consider another terminology.*

The term "complete digital karyotype" has been reworded to "detailed genomic and transcriptomic characterization" per the reviewer's earlier suggestion.

- 8. Page 5 The following abbreviations are not defined: FLNC, CLR and CCS.*

These abbreviations have now been defined as "full-length non-chimeric reads", "continuous long read", and "circular consensus sequencing", respectively.

9. Page 9 "We used G-banding to confirm...." G-banding is a method for staining chromosomes. Not an SV analysis technique. G-banded karyotyping is the analysis technique.

We replaced all instances of "G-banding" with either "G-banded karyotyping", or simply "karyotyping", where the context was clear.

10. In the supplemental materials referred to on page 9 - Supplemental Table S2 this reviewer notes several errors in the ISCN.

From S2

Authors Version

46, X, -X, dup(1)(p21.1), t(1;2)(q21.1;p11.2)-inv(2)(p11.2), del(3)(p14.2p22.3), del(3)(q26.1), del(3)(q26.32), t(4;12;21;16)(q32.1;p13.2;q22.12;q24.3)-inv(12)(p13.2q23.1), del(5)(q31.3), t(5;12)(q23.2-q23.3;p13.2), del(6)(q21), del(12)(q21.33), del(14)(q24.2), inv(16)(p12.2), der(16)t(16;21)(q24.3;q22.12) x 2, del(18)(q21.1), del(18)(q23) x 2, del(22)(q11.22) x 2

Reviewers Suggestion

46,X,-X,dup(1)(p21.1p21.1),der(1)t(1;2)(q21.1;p11.2),der(2)t(1;2)inv(2)(p11.2p.11.2),del(3)(22.3p14.2),del(3)(q26.1),del(3)(q26.32q26.32),der(4)t(4;12;21;16)(q32.1;p13.2;q22.12;q24.3),del(5)(q31.3q31.3),t(5;12)(q23.2-q23.3;p13.2),del(6)(q21q21),der(12)t(4;12;21;16)inv(12)(p13.2q23.1)del(12)(q21.33),del(14)(q24.2q24.2),inv(16)(p12.2p12.2),der(16)t(4;12;21;16)x2,del(18)(q21.1q21.1),del(18)(q23q23)x2,del(22)(q11.22q11.22)x2

We thank the reviewer for their attention to detail in reviewing the ISCN notation, and for bringing to our attention the problems therein. After consultations with our cytogeneticist, co-author Anna Bremer, we have come to realize that it is not possible to accurately convey the entirety of our findings using ISCN 2020 notation. Indeed, the conventional ISCN notation intended for g-banded karyotyping can only accurately describe re-arrangements that span more than one cytogenetic band. For smaller aberrations detected using genomic sequencing, ISCN 2020 introduces "Sequence-Based Nomenclature" (Chapter 16). However, this chapter does not indicate the proper notation for the copy-neutral loss of heterozygosity that we have now discovered in the p-arm of chromosome 9 (see response to Reviewer #2, point 4 below). To denote this aberration, one would have to use the symbol "hmz" (Chapter 14.2.6), but this notation is intended for microarrays, which were not used in this project. Further, ISCN cannot be used to denote the SNVs which we have now characterized in the manuscript (see response to Reviewer #2, point 3 below). Therefore the original idea of creating a single "digital karyotype" integrating all of our results becomes challenging. In light of these limitations, and the fact that cytogenetic notation is a rapidly changing field, with a new version ISCN 2024 forthcoming, we have chosen to remove the cytogenetic notation altogether from the results in the main manuscript.

However, we have retained its presence in Supplementary Table S1, with the intent to compare only the largest aberrations with the available g-banded karyotypes. The table is now titled "Table S1. Comparison of REH chromosomal features > 5 Mb across karyotypes", and contains the entry "Lysenkova et al sequence-based karyotype" as follows:

46,X,-X,del(3)(p22.3p14.2),der(4)t(4;12;21;16)(q32.1;p13.2;q22.12;q24.3),t(5;12)(q23.2-q23.3;p13.2),der(12)t(4;12;21;16)inv(12)(p13.2q23.1),der(16)t(4;12;21;16)x2,der(21)t(4;12;21;16)

This incorporates the reviewer's suggestions with some modifications, which we detail in the sub-points below.

Detailed list of changes to ISCN with justification.

1. *all spaces removed.*
Change accepted.
2. *dup(1)(p21.1p21.1) is correct for a small segmental duplication.*
Removed from the "Lysenkova et al sequence-based karyotype" due to small size (< 5 Mb).
3. *t(1;2)(q21.1;p11.2)-inv(2)(p11.2). This is not a defined nomenclature style in the ISCN. der(1)t(1;2)(q21.1;p11.2),der(2)t(1;2)inv(2)(p11.2p.11.2) is a better possibility. If the inv(2)(p11.2p11.2) is closer to the p-terminal than the t(1;2) breakpoint it should be listed first, e.g. der(2)inv(2)(p11.2p.11.2)t(1;2). This nomenclature is correct only for a balanced translocation. If translocation is unbalanced, as suggested by authors, needs to be clarified what are the gains or losses to correctly write nomenclature. I could not ascertain this from the data presented. N.B. detailed breakpoints on the t(1;2) not required as they were defined on the der(1). **see additional points regarding page 12 on this rearrangement - need to be clarified 1) exact mechanism 2) fig 5 3) ISCN*
This has been corrected to the following:
der(1)inv(2)(p11.2p11.2)ins(1;2)(q21.1;p11.2) with the reasoning detailed in point 19 below. While this aberration has been removed from the "Lysenkova et al sequence-based karyotype" due to small size (< 5 Mb), we use this updated notation to refer to the aberration in the manuscript. Figure 5 has been updated accordingly:

Figure 5 - The aberrant chromosomes of the REH cell line. Highlighted are the retained protein domains for ETV6::RUNX1 and RUNX1::PRDM7, the two REH in-frame fusion genes. Rearrangements that could not be phased to a specific homolog have been rendered on an arbitrary homolog of their respective chromosomes.

4. *del(3)(p14.2p22.3)* should be *del(3)(p22.3q14.2)* breakpoints are always from pter to qter in the ISCN.

This has been corrected to the following: *del(3)(p22.3p14.2)*, as the second breakpoint is at p14.2, not q14.2. We have also updated the notation for this aberration used in the manuscript.

5. *del(3)(q26.1)* - should be *del(3)(q26.1q26.1)* per fig 5.

Removed due its identification as a germline variant (see response to Reviewer #2, point 3 below).

6. Same point above for *del(3)(q26.32)*

While this aberration has been removed from the “Lysenkova et al sequence-based karyotype” due to small size (< 5 Mb), we use the updated notation *del(3)(q26.32q26.32)* to refer to the aberration in the manuscript.

7. For the *t(4;12;21;16)* since the 12 has a compound abnormality it cannot be described without using *der*. I have expanded the nomenclature above.

Change accepted, with one modification: since *der* chromosomes are suggested for chromosomes 4, 12, and 16, we have also specified *der(21)* for consistency.

8. By karyotype (figure) the *del(5)(q31.3)* does not appear to be a terminal deletion as that would be visible by karyotype. I have corrected the ISCN to read as a segmental deletion on chromosome 5 as I am assuming this is correct (also per Fig 5). Also, when both chromosome homologues are involved in rearrangements it is often beneficial to differentiate one from the other by underlining the chromosome number when alternate homologues are involved in different rearrangements. For example, the *t(5;12)* involves one homologue, but the *del(5q)* is on the other and so the “5” can be underlined for clarity. Apologies for not showing this however the text editing capabilities of the review platform do not allow. Please see any relatively recent version of the ISCN for detailed instructions - underlining (single) Used to distinguish homologous chromosomes (ISCN sections 4.1, 9.2.3, 9.2.17.1. <https://iscn.karger.com/Home/Overview> N.B. while ISCN 2020 requires a subscription previous version can be downloaded as PDFs

This deletion has been removed from the “Lysenkova et al sequence-based karyotype” due to small size (< 5 Mb), making the underlining unnecessary; the notation has been updated to *del(5)(q31.3q31.3)* when referring to this aberration in the manuscript. It should also be noted that due to the current limitations of WGS, even with long reads, it is often impossible to know whether two rearrangements occur on the same or different homologs of a single chromosome when both g-banding and phasing analysis fail to resolve such ambiguity. The legend for Figure 5 has been updated to reflect this ambiguity: “Rearrangements that could not be phased to a specific homolog have been rendered on an arbitrary homolog of their respective chromosomes.”

9. *del(6)(q21)* changed to *del(6)(q21q21)* to represent a segmental deletion as opposed to a terminal deletion.

While this aberration has been removed from the “Lysenkova et al

sequence-based karyotype” due to small size (< 5 Mb), we use this updated notation to refer to the aberration in the manuscript.

10. *By fig 5 the del(12q) a part of the der(12) with the t(4;12;21;16). It can be added to the der(12) nomenclature*

der(12)t(4;12;21;16)inv(12)(p13.2q23.1)del(12)(q21.33).

Per point 10.8 above, it is not feasible to confirm which derivative chromosome this del(12q) occurs on. Since it is not a terminal deletion, we have revised it to del(12)(q21.33q21.33) in the manuscript, and removed it from the “Lysenkova et al sequence-based karyotype” due to small size (< 5 Mb).

11. *del(14)(q24.2) changed to del(14)(q24.2q24.2) assumed, as above, segmental as loss of 1/3 for 14q would be visible.*

Removed from the “Lysenkova et al sequence-based karyotype” due to small size (< 5 Mb).

12. *inv(16)(p12.2) changed to inv(16)(p12.2p12.2) to indicate an inversion within the same band. May wish to underline this if known on other homologue.*

Removed from the “Lysenkova et al sequence-based karyotype” due to small size (< 5 Mb).

13. *der(16)t(16;21)(q24.3;q22.12)x2 changed to der(16)t(4;12;21;16)x2 to indicate to copies of the der(16).*

Change accepted.

14. *del(18)'s and del(22) nomenclature corrected by comparison with Figure 1 and Figure 5.*

Removed from the “Lysenkova et al sequence-based karyotype” due to small size (< 5 Mb); references in the manuscript have been updated.

11. *Page 10 "a 26mbp del(3)(p22.3;p14.2)" should read "26 Mb del(3)(p22.3p14.2)*

This has been changed according to the reviewer's suggestion.

12. *"a complex four-way rearrangement between chromosomes 4, 12, 21, and 16, resulting in the ETV6-RUNX1 fusion gene" refer to my previous point about complex. Also ETV6-RUNX1 should be ETV6::RUNX1.*

Per point 3 above, the word “complex” has been removed. All instances of “ETV6-RUNX1” have been replaced with “ETV6::RUNX1”; additionally, all other fusions have been renamed using this double-colon convention in the main manuscript, supplementary, Table 2, and Figure 5.

13. *"involve two different alleles of chromosome 12" I would suggest using the term "homologue" instead of allele. Allele refers to two version of gene. In this case you are saying that the t(5;12) and the t(4;12;21;16) involve both chromosome 12 homologues.*

We have implemented this change; however, in keeping with the standard of US English throughout our manuscript, we have opted for the spelling “homolog”.

14. "as the two different p-arms of chromosome 12 (from p13.2 to the telomere)" - would suggest p-terminal instead of telomere.

This change has been implemented.

15. "The p-arm of chromosome 12 (12Mbp), with breakpoint at p13.2 in intron 5 of ETV6, was translocated to chr21q22.12, resulting in the canonical, subtype-defining fusion gene ETV6-RUNX1." - I would suggest "The p-arm of chromosome 12, with breakpoint at p13.2 in intron 5 of ETV6, was translocated to chr 21q22.12, resulting in the canonical, subtype-defining fusion gene ETV6::RUNX1."

Per point 12 above, ETV6::RUNX1 and all other fusion genes have been updated to use the double-colon convention.

16. Page 11 "completing the circular exchange." - suggest 'completing the reciprocal exchange'.

This change has been implemented.

17. "While this evidence supports the presence of two copies of der(16)t(16;21)" - with the updated ISCN you can now refer to this as simply the "der(16)" and not use the incorrect nomenclature t(16;21).

We have changed this in both the main manuscript and supplementary.

18. "were previously unidentified in the analog karyotypes" - the term "analog" seems out of place. I would remove all instances of the term 'analog karyotype(s)' in favour of simply 'karyotype(s)'.

The term "analog" has been removed entirely from the manuscript. As "digital karyotype" has been removed and replaced by terms such as "digital characterization", it can now be inferred that all instances of the word "karyotype" refer to the product of the conventional g-banding technique.

19. Page 12 "We identified a pair of rearrangements on chromosome 2 including a 1.4 Mbp inversion at p11.2 and an adjacent, unbalanced 58 kb translocation resulting in derived chromosome der(1)t(1;2)(q21.1;p11.2)." An unbalanced translocation would be associated with a huge region of copy number imbalance. If there is no detected copy number imbalance is this because the t(1;2) is at low VAF or is it possibly an insertion?

This was an error on our part. The term "unbalanced translocation" was incorrectly used to describe what is, in fact, the insertion of an inverted segment of chromosome 2 into chromosome 1. According to ISCN 2020, the correct nomenclature for this variant should

be: $der(1)inv(2)(p11.2p11.2)ins(1;2)(q21.1;p11.2)$. Since the breakpoint for the inversion on chromosome 2 is closer to the p-terminus than the breakpoint for the insertion, the inversion is listed first, per the reviewer's comment in point 10.3 above. The passage in question has been revised to the following: "*A rearrangement was identified involving a 1.4 Mb segment of chromosome 2 inverted at p11.2, 58 kb of which was inserted into chromosome 1, resulting in derived chromosome $der(1)inv(2)(p11.2p11.2)ins(1;2)(q21.1;p11.2)$.*" Other references to this rearrangement have been updated in both the main manuscript and supplementary.

20. *Figure 5 does not show a translocation $t(1;2)$. Also, the karyotype in Figure 1 does not show a $t(1;2)$. Is this translocation a sideline? Or with the presence of a deletion is it possible that it is in fact an insertion? Some comment needs to be made regarding why the $t(1;2)$ is not shown on Fig 5 or the figure corrected.*

Per point 19 above, the variant in question is in fact an $ins(1;2)$. Figure 5 has been updated with this correction.

21. *Page 14 Comprehensive karyotype - please update the nomenclature in this section.*

This section has now been integrated with SNV discovery (see response to Reviewer #2, point 3 below), so the title of this section has been updated to: "Key SNVs conclude the detailed characterization of the REH cell line." The final passage also removes "comprehensive karyotype" language in favor of the following: "*Combining cytogenetics, SV analysis and SNV detection, our integrative short- and long-read genomic and transcriptomic analysis allows us to provide a detailed characterization of the REH cell line, shown in Figure 5.*"

Comments from Reviewer #2

Major points:

1. *I'm afraid that the detected SVs are not "comprehensive". Not all the SNVs residing in this cell line is not represented. Or at least, the precise evaluation on the precision and recall rates are not validated, perhaps using other well-characterized material. More practically, I do not think it is always good idea to pick up the SVs which were detected from PacBio/ONT and Illumina in a (three-way) overlapping manner. For example, the largest advantage in using the long reads should lie in the cases which the short reads have overlooked, but this population would be missed when the overlapping population was picked. However, on the other hand, the data would contain substantial errors if all the results solely based on the long reads were picked up without any filtration. Therefore, a more careful filtration strategy should be considered.*

We agree that the term "comprehensive" is not appropriate. Per our response to point 1 from Reviewer #1, above, we have removed the use of this word throughout the

manuscript in favor of terms such as “detailed”.

We also agree that the three-way consensus set is not an optimal source of truth, as taking the intersection of the short- and long-read callsets excludes the SVs called by long-read sets only, and this fails to take advantage of the SV detection benefits uniquely provided by long reads.

For this reason, the consensus sets were not used for the detection of the large scale rearrangements (> 100 kb) detailed in Table 2 and used as the basis for the final schematic in Figure 5. Instead, for these rearrangements, the three unmerged SV callsets were filtered and inspected separately. This methodology is detailed in the Supplementary section “3.3 Filtering of SV candidates”. We have also clarified this in the main manuscript section “The SV landscape stratified by size”, by specifying that “*The original Sniffles and TIDDIT callsets were then programmatically filtered (see Materials and Methods).*”

Upon further reflection on the reviewer’s point, we made further revisions to the section “The SV landscape stratified by size”. We created a new long-read consensus dataset, which contains the overlapping set of all SVs detected in both the PacBio and ONT datasets, but not the Illumina data. Such a dataset leverages the unique capabilities of long reads to resolve SVs, while mitigating the risk of false positives as a result of vendor-specific artifacts by including only those SVs where both long-read datasets agree. The Github repository has been updated with the relevant code.

Figure 2 has been adjusted to reflect this more informative dataset. Figure 2A, a chromosomal heatmap of SV calls, now visualizes this long-read consensus callset instead of the 3x consensus callset, as it did previously.

Figure 2 - REH structural variants detected in PacBio, ONT and Illumina WGS. (A) chromosomal heatmap of the long-read consensus callset, showing the total number of SV calls at each locus that were detected in both PacBio and ONT data

Similar heatmaps of the 3x consensus set, and of the individual unmerged callsets, have been added to Supplementary as Figure S4.

In panel C of Figure 2, we have updated the strip plots to show both the 3x consensus set and the new long-read consensus set, instead of just the three-way consensus set as previously:

Figure 2 - REH structural variants detected in PacBio, ONT and Illumina WGS. (C) strip plots showing SVs from each callset, stratified by size, with the bottom two strips in each section visualizing the long-read consensus callset and the 3-way consensus callset, respectively.

2. *Statistics of the sequencings are not detailed. Especially, the read length and the sequencing fidelity of ONT are important. It is potentially advantageous that the ultra-long reads of ONT were employed in this study, similar to the T2T project. However, it is not clear how this advantages are utilized.*

We agree that this information is highly relevant for this manuscript. A report containing key statistics, including sequencing error rates, for the three WGS and two RNA-seq datasets has been generated by the Alfred BAM Statistics tool and provided as Supplementary Table S2.

The read length and error rates have also been added to Table 1 and are summarized in the following passage from the first Results section, “Overview of the genome”: “*The REH cells were subjected to genomic and transcriptomic sequencing. Of 33 datasets generated (Lysenkova Wiklander et al, 2023), the present study used three WGS and two RNA-seq datasets (Table 1). The Illumina WGS dataset had an average depth of coverage of 34 reads, while ONT had 18 and PacBio 15; the median read lengths were 152 bp, 13261 bp and 23761 bp, respectively (Figure 1B). The IsoSeq median read length was 3747 bp (Supplementary Figure S1), while the Illumina RNA-seq median*

read length was 102 bp. All sequencing datasets had an error rate of less than one percent, with the exception of ONT (6.55%). Further sequencing statistics of these datasets are detailed in Supplementary Table S2.”

We have added a panel showing the read length distribution in the ONT and PacBio datasets to Figure 1:

Figure 1 - Overview of the REH cell line. (B) The read length distributions of the long-read WGS datasets.

The advantages of long reads were previously detailed only in the supplementary material, but we now summarize it in the results section “Large-scale SVs discovered in the REH genome”: “Overall, Sniffles run on ultralong ONT reads outperformed in the discovery of large-scale rearrangements, showing high sensitivity (95.65%) and a low false-positive rate relative to the other datasets (85.33%) (Supplementary Figure S10; Supplementary Table S4; Supplementary Results 1.4).” In the supplementary section “1.4 Long-read WGS outperformed for translocation discovery and resolving highly repetitive regions”, we go into further detail.

We also address the advantages of long reads in the following passage from the discussion: “In assessing the performance of the long- and short-read technologies, we

found benefit in integrating a variety of tools and datasets with distinct strengths in detecting variants of different types and sizes (van Belzen et al, 2021). Overall, we found that the ONT dataset was the most useful for SV discovery, with the highest sensitivity and lowest FPR; however, when adjusting for the slightly lower DC, PacBio had a comparable performance. Importantly, both long-read datasets were able to resolve variants in highly repetitive regions that were missed in the short-read data."

We now also address the drawbacks of long reads in the discussion:

"Our long-read datasets had relatively lower coverage (15x PacBio, 18x ONT) than the Illumina data set (34x). Additionally, the ONT sequencing data was generated using a version R9.4.1 flow cell, resulting in a mismatch rate of 3% and overall error rate of 6.55%, which is in line with the reported error rates for this ONT flow cell version, but was higher than the error rates for the other data sets (<1%)."

- 3. In addition to SVs, SNVs should be also characterized (or validated if this part is previously characterized, perhaps in CCLE). Also, as the normal counterpart of this cell line is missing, at least, some efforts should be made to exclude its germline variations, both for SVs and SNVs.*

We performed SNV analysis using the following methodology, which we have now detailed in the final section of results, under the heading "Key SNVs conclude the detailed characterization of the REH cell line": *"Finally, we characterized the SNVs in the REH genome. For SNV discovery, only the Illumina data was used, due to the higher coverage and lower error rate of this data set in comparison to the long-read datasets (Supplementary Table S2). Key SNVs, detailed below, were subsequently corroborated in the long-read data.*

*Mutect2 was run on the Illumina WGS dataset, filtered against a panel of normals (PON) from the 1000 Genomes Project and allele frequencies from gnomAD, to produce a set of SNVs that were further filtered to include only variants affected by loss of function (LOF) or nonsense-mediated mRNA decay (NMD). The resulting set contained 131 variants (Supplemental Table S8), including a nonsense mutation in exon 5 of GC receptor NR3C1 (p.Gln528Ter; Supplemental Figure S11) and a frameshift mutation in exon 8 of PAX5 (p.A322Rfs*19; Supplemental Figure S12). Phasing analysis placed the NR3C1 mutation on the same homolog of chromosome 5 (contig 56) as the t(5;12)."*

For the large-scale SV's, we manually checked them against gnomAD SVs v2.1. Indeed, the 114 kb deletion at chr3q26.1 was found to be a germline variant (http://www.gnomad-sg.org/variant/DEL_3_39037?dataset=gnomad_sv_r2_1) and has now been excluded from our results, including Table 2, all SV statistics, and Figures 3 and 5.

Minor points:

4. For identifying aneuploidy regions, particularly careful evaluation should be needed. The loss of whole X chromosome may be the easiest case. However, a wide variety of other factors, which may affect in combination, can be assumed (for example, duplication + loss= seemingly diploid is a very common event). For this purpose, the ration of heterozygous SNPs should give a useful information.

We have now performed an additional Variant Allele Frequency (VAF) analysis using SNVs.

This analysis has uncovered copy number neutral loss of heterozygosity across much of the p-arm of chromosome 9, which we detail in the first results section, "Overview of the genome": "Three sets of single nucleotide variant (SNV) calls were generated by running DeepVariant on the WGS datasets. Variant allele frequency (VAF) of the SNVs were used to assess uniparental disomies and other potential losses of heterozygosity (LOH) in the cell line (Figure 1C; Supplementary Figure 2). In addition to confirming the del(3), +16 and -X, a partial trisomy of chromosome 21 was detected, suggesting two copies of the der(16) chromosome resulting from the t(4;12;21;16). Copy neutral loss of heterozygosity (cnLOH) was also detected across much of the short arm of chromosome 9, stretching from p13.2 (between exons 4 and 5 of RNF38) to the p-terminal."

Figure 1 has been updated with the resulting plot from Illumina, which had the best quality. In addition, the corresponding plots from the long-read datasets are available in Supplementary Figure 2.

Figure 1 - Overview of the REH cell line. (C) The variant allele frequencies (VAFs) of the single-nucleotide variants called by DeepVariant in the Illumina WGS data. The allele fractions of these SNVs, relative to the reference alleles, are binomially distributed, with 1.0 indicating homozygous variants and a mean of 0.5

indicating heterozygous variants.

5. *Similarly, chromosomal context of each of the SVs can be resolved by further inspecting their surrounding SNPs, taking advantages of the long read (by "phasing" analysis).*

We performed phasing analysis to resolve the chromosomal context of the larger SVs detailed in Table 2. However, this has been quite tricky. Conventional phasing analysis does not provide phasing blocks long enough to assess the relationship between these SVs, most of which are separated by 1 Mbp or more. An alternative approach, which we pursued, is de-novo assembly, which is capable of generating contigs that are multiple megabasepairs in length.

However, this approach had significant shortcomings - thus, the results are in supplementary only, where in section "1.5 Phasing of the large-scale SVs", we detail the limitations: "*Phasing of the large-scale SVs was performed by creating a de-novo assembly of the ONT reads with Flye and haplotyping with Hapdup, followed by alignment of the haplotypes to the reference genome and SV calling using Hapdiff. One can infer whether a pair of SVs occur on the same homolog if they occur on the same contig or, alternatively, if a single contig covers the breakpoints of both events and contains neither of the SVs. The three large-scale deletions on the q-arm of chromosome 18 were mapped to a single contig, indicating that they all occur on the same homolog. However, this approach struggled to properly resolve rearrangements with higher complexity; for example, it mapped the large-scale inv(12) to the same homolog as the t(5;12), contradicting cytogenetic evidence. Further, of the 31 SV breakpoints that were assigned to a haplotype, 14 of the contigs containing heterozygous SVs (45%) were erroneously assigned to both maternal and paternal haplotypes, while Hapdiff only called 10 of the 24 confirmed large-scale SVs (Supplementary Table S5).*"

We further clarify these limitations up in the discussion: "*While this approach leveraged the ability of long reads to assemble into multi-megabasepair contigs, potentially allowing the resolution of chromosomal context for distant SVs, it suffered from misassembly and difficulty properly resolving haplotypes. It has been previously noted that de-novo assembly in an oncological context may struggle to construct accurate contigs due to the complexity and heterozygosity of cancer genomes (Sakamoto et al, 2021), although where matched tumor-normal samples are available, sequencing and de-novo assembly of both samples, followed by a comparative analysis, can be used for accurate, personalized SV detection (Xiao et al, 2022).*"

6. *For the possible "fusion" transcripts, do they likely to have any functional relevance, judging from the transcriptome data, which also represents the gene expression pattern?*

Of the fusions confirmed in the manuscript, the following have functional implications:

- a. *ETV6::RUNX1* (in-frame): this subtype-defining fusion has been extensively documented in ALL literature and in the present manuscript.
- b. *RUNX1::PRDM7* (in-frame): the functional relevance of this fusion gene is more speculative, but we do address its potential implications in the following passage from the discussion: “*PRDM9, which shares the SSXRD, SET and KRAB protein domains with PRDM7, has been implicated in genetic predisposition to pediatric B-ALL (Hussin et al, 2013; Thibault-Sennett et al, 2018), while our findings suggest that this association might extend to other members of the PRDM gene family.*”
- c. *BTG1::LINC02404/AC090049.17*: the fusion results from the truncation of *BTG1* at exon2; the disruption of this gene has been shown to contribute to glucocorticoid resistance, as noted in the discussion: “*The REH cell line comes from a relapsed patient, making it particularly notable that several deletions identified in the present study, affecting genes TBL1XR1 (146k deletion on 3q26.32), NR3C1 (205 kb deletion on 5q31.3) and BTG1 (260 kb deletion on 12q21.33), have been shown to contribute to its GC drug resistance (van Galen et al, 2010; Jones et al, 2014; Xiao et al, 2019).*”

The remaining fusion genes were lowly expressed (detailed in Supplementary Table S6) and were not in-frame, but their relevance cannot be ruled out. The present study is intended to be only a preliminary step in investigating whether the genomic aberrations detailed therein have functional relevance. Only if these aberrations are detected in patients can any conclusions be definitively made.

7. *I wonder if there is any sign for possible emergence of a minor sub-population? The cell line may not be stable after long history of cell culture.*

This is an excellent point. The REH cells used for sequencing in this project have now been subjected to STR analysis (with Eurofins human STR profiling cell line authentication services) to verify the authenticity of the cell line. We have also performed an additional analysis which we have described in the “Overview of the genome” section of the results: “*To rule out the presence of subclones, the allele fractions of the three-way consensus callset, containing the SVs called in all three WGS datasets, were plotted to ensure that they follow a binomial distribution (Supplementary Figure S3).*”

Supplementary Figure S3 - Distribution of allele fractions of the REH structural variants in the three-way consensus callset. The three-way consensus callset contains the SVs called in all three WGS datasets (Illumina, PacBio and ONT). The allele fractions of these SVs are binomially distributed, with 1.0 indicating homozygous variants and a mean of 0.5 indicating heterozygous variants.

8. *I'd like to suggest that DNA methylation call is also possible, which may render this dataset represent more multi-omics features.*

We agree that methylation calling of the REH cell line would be an interesting contribution; in particular, it would be interesting to analyze the methylation patterns in the regions surrounding the SVs in this genome. However, we believe that this analysis is better suited to a new study. Firstly, there is the question of scope, as epigenomic analysis would significantly increase the complexity of the present study, which is already quite long. Secondly and more importantly, newer datasets would be more suitable for this analysis. The long-read whole genome sequencing (WGS) datasets analyzed in this study were generated in 2021 (PacBio) and 2022 (ONT). Methylation calling on PacBio data generated at that time is not possible, and although methylation calling on the ONT dataset is feasible, recent advances in long-read technology from both companies have significantly improved this process. Additionally, hydroxymethylation calling is now possible¹, potentially adding a new facet to epigenomic analysis. We appreciate this suggestion, and as we plan to continue using the REH cell line for testing new and cutting-edge long-read technologies, we will have the opportunity to explore this cell line's epigenome.

¹ <https://doi.org/10.1186/s13059-020-01969-6>

Other changes

- General:
 - The sections of the manuscript have been reordered to reflect author guidelines
 - Stylistic edits have been made throughout the manuscript
 - Software versions and links have been removed from the manuscript for readability, and are detailed instead in Supplementary Table S9
- Title page:
 - The author affiliations have been revised to adhere to the latest guidelines from Uppsala University
 - A running title has been added
 - Keywords have been added
- Abstract has been revised to adhere to the maximum word count in the author guidelines
- Graphical abstract has been included
- A summary blurb has been added
- Discussion: a paragraph has been added detailing the limitations of the present study, combining several points brought up by the reviewers
- Acknowledgments:
 - Added grant details for computing resources
 - Acknowledged BioRender.com
- Data Availability:
 - Zenodo links have been updated to the DOI reflecting the latest version
- References:
 - have been reformatted according to author guidelines
 - references to pre-prints that have been published since the previous revision have been updated
- Supplementary:
 - In keeping with LSA's style of providing supplementary figures and tables inline, the Supplementary Information PDF has been removed in favor of Supplementary Information at the end of the main manuscript file
 - All instances of "Supplemental" have been replaced with "Supplementary".
 - Table S1: the three datasets whose submission was previously in progress have been published and their NCBI/SRA accession numbers added to the table.
 - One of the original de-novo assemblies (Flye polished with Racon) was found to have been generated incorrectly and has been removed from the manuscript.

Sincerely,
Jessica Nordlund,
Corresponding author

April 25, 2024

RE: Life Science Alliance Manuscript #LSA-2023-02481R

Dr. Jessica Nordlund
Uppsala University
Medical Sciences
BMC, Box 1432
Uppsala 75144
Sweden

Dear Dr. Nordlund,

Thank you for submitting your revised manuscript entitled "A multiomic characterization of the leukemia cell line REH using short- and long-read sequencing". We would be happy to publish your paper in Life Science Alliance pending final revisions necessary to meet our formatting guidelines.

- please be sure that the authorship listing and order is correct
- please add the Twitter handle of your host institute/organization as well as your own or/and one of the authors in our system
- please add callouts for Figures 2A-C; S2A-B; S4A-D and S10A-C to your main manuscript text
- the Supplementary Results, Methods and Discussion text should be incorporated into the main manuscript. Results can refer to Supplemental Figures

A. FINAL FILES:

B. MANUSCRIPT ORGANIZATION AND FORMATTING:

**Submission of a paper that does not conform to Life Science Alliance guidelines will delay the acceptance of your

manuscript.**

The license to publish form must be signed before your manuscript can be sent to production. A link to the electronic license to publish form will be available to the corresponding author only. Please take a moment to check your funder requirements.

Thank you for your attention to these final processing requirements. Please revise and format the manuscript and upload materials within 4 days.

Sincerely,

Reviewer #1 (Comments to the Authors (Required)):

This resubmitted manuscript has addressed many of this reviews concerns making the manuscript much stronger overall.

There is also a considerable amount of data that is presented on this single cell line demonstrating the complexity of this type of analysis.

The response to reviewers suggests that "Indeed, the conventional ISCN notation intended for g-banded karyotyping can only accurately describe re-arrangements that span more than one cytogenetic band." I believe this to be incorrect as a deletion within a single band or an insertion (with both breaks in the same band) could still be described using conventional ISCN. However, this is a minor point. The sequencing based nomenclature could be adapted to describe the smaller SV events in the paper - however, this reviewer also realizes this may be a herculean task in the presence of many small SVs. Therefore, the choice to limit the description of many of the findings to >5Mb is understandable. Also, while an example may not exist in the sequencing based nomenclature for LOH detection "hmz", the ISCN functions as guidance more than a strict rule book. However, this reviewer also hopes the 2024 ISCN will provide expanded examples for sequencing based description of SVs. Also, the sequencing based nomenclature does also provide a mechanism to report SNVs co-occurring with SVs. However, providing an excessively long string of SVs and SNVs is likely not the most efficient mechanism for cataloguing these results. ISCN 2024 will include some guidance on using tables for SV reporting - which may help with very complex genomes.

The authors acknowledge in their response one of the big challenges of this field moving forward and that is how to accurately and consistently provide nomenclature for all the SVs that are detected. I concur with this statement and feel that contributions to the literature, like this manuscript, should help to demonstrate the need for improved nomenclature methods and/or systems.

In conclusion, the resubmitted manuscript has effectively addressed my concerns and I have no further comments.

Reviewer #2 (Comments to the Authors (Required)):

First of all, I appreciate the substantial efforts of the authors made for this revision. Thanks to the extensive analyses and detailed descriptions, I believe this manuscript has been very much improved. Although there are still remaining analyses, which should be included before this dataset can be truly regarded as "comprehensive", I admit that part should be subjected to their future research. After all, I think the benefits should be larger than the shortcomings at this stage. As I notice that the authors fully realize this issue from their careful discussion, I believe they should further improve the current dataset.

May 2, 2024

RE: Life Science Alliance Manuscript #LSA-2023-02481RR

Dr. Jessica Nordlund
Uppsala University
Medical Sciences
BMC, Box 1432
Uppsala 75144
Sweden

Dear Dr. Nordlund,

Thank you for submitting your Resource entitled "A multiomic characterization of the leukemia cell line REH using short- and long-read sequencing". It is a pleasure to let you know that your manuscript is now accepted for publication in Life Science Alliance. Congratulations on this interesting work.

DISTRIBUTION OF MATERIALS:

Again, congratulations on a very nice paper. I hope you found the review process to be constructive and are pleased with how the manuscript was handled editorially. We look forward to future exciting submissions from your lab.

Sincerely,
